



# Evaluating the Impact of Motion Compensation on Turbulence Intensity Measurements from Continuous-Wave and Pulsed Floating Lidars

Warren Watson[a], Gerrit Wolken-Möhlmann[a], and Julia Gottschall[a]

[a]Fraunhofer Institute for Wind Energy Systems IWES, Am Seedeich 45, 27572 Bremerhaven, Germany

**Correspondence:** Warren Watson (warren.watson@iwes.fraunhofer.de)

**Abstract.** Floating Lidar Systems (FLS) play a crucial role in offshore wind resource assessment, offering a cost-effective and flexible alternative to traditional meteorological masts. While wind speed and direction measurements from FLS demonstrate high accuracy without further in-depth correction required, platform motions introduce systematic overestimation of turbulence intensity (TI). This motion-induced bias requires compensation techniques to ensure reliable TI measurements. This study
5    evaluates the impact of a deterministic motion compensation algorithm on TI measurements from two FLS of the same type, equipped with different lidar types: a continuous-wave (cw) lidar and a pulsed lidar. The analysis compares raw and motion-compensated TI data against reference measurements from a fixed cw lidar and a met mast cup anemometer.

A comprehensive evaluation is conducted using multiple performance metrics, including Regression Analysis, Mean Bias Error (MBE), Mean Relative Bias Error (MRBE), Root Mean Square Error (RMSE), Relative Root Mean Square Error (RRMSE),
10    Representative TI Error, and Quantile-based distribution analysis. The results show that the applied motion compensation significantly reduces the overestimation of TI, with the pulsed lidar exhibiting the most substantial relative improvement across various metrics. The cw lidar, while also benefiting from motion compensation, demonstrates a closer alignment with the fixed lidar in terms of absolute bias reduction.

Despite these improvements, residual discrepancies remain, attributed to differences in measurement principles, remaining
15    motion effects, lidar-specific characteristics and sensitivities. Our findings confirm that deterministic motion compensation can enhance the reliability of FLS-derived TI measurements, bringing them closer to those obtained from a fixed lidar system. Future work should focus on refining compensation algorithms by incorporating lidar-specific sensitivities, improving sensor time synchronization, and exploring machine learning-based enhancements for an even better agreement with a met mast reference.



## 1 Introduction

Offshore wind energy has gained significant momentum as a key component of the global transition to renewable energy (WindEurope (2024)). Accurate site assessment is crucial for the successful development of offshore wind farms, as it directly impacts the feasibility, efficiency, and economic viability of these projects. Advanced wind measurement technologies, including meteorological masts and remote sensing devices, play a pivotal role in understanding wind conditions over open water (Sempreviva et al. (2008)).

Floating lidar systems (FLS) represent an innovative advancement over meteorological masts in offshore wind resource assessment (Gottschall et al. (2017)). These systems are designed to provide accurate and reliable measurements of wind speed, direction, and other meteorological and oceanographic parameters in harsh offshore environments where traditional meteorological masts may be difficult to deploy and maintain. FLS typically consist of one or more vertical profiling wind lidars mounted on or integrated in a floating platform. These systems are designed to operate autonomously, featuring self-sufficient power supplies, multiple communication systems, motion recording devices (inertial measurement units - IMU), and various supplementary measurement instruments.

The OWA Roadmap to the Commercial Acceptance of Floating LiDAR Technology (OWA (2018), also referred to as OWA Roadmap) defines a structured approach to prove the accuracy, quality as well as reliability of the individual FLS system within three stages of maturity. Herein defined Key Performance Indicators (KPIs) focus on wind speed and wind direction data. While the upcoming IEC 61400-50-4 technical specification (IEC (2024)) is expected to provide further guidelines for the classification, calibration, and application of FLS, the OWA Roadmap currently remains a key reference for their commercial acceptance. In recent years, several FLS types have reached the full commercial maturity stage (Stage 3: Commercial Stage), underlining the capability of FLS in terms of wind speed and direction measurement accuracy, as well as system and data availability.

In contrast to wind speed and wind direction accuracy, FLS are known to overestimate the turbulence intensity (TI) compared to an unmoved, fixed lidar. TI is defined as the ratio of the standard deviation of the wind speed to the mean wind speed and is a key parameter for characterizing atmospheric turbulence in the wind energy context. This overestimation occurs because lidars mounted on floating platforms are subject to wave-induced movements. As the platform moves, the lidar device itself experiences motion, which introduces fluctuations in the measured wind speed and consequently leads to an increased standard deviation. The extent of this effect depends on the prevailing sea state, the specific design characteristics of the FLS including its dynamic response to sea-state conditions, as well as the deployed lidar type and its configuration. As these sea state-induced motions are mainly periodic, their effects on the 10-minute average wind speed are small, while the TI and wind direction are significantly influenced (Gottschall et al. (2014)). Different FLS types exhibit varying levels of sensitivity to these motion-induced effects, depending on their buoy design, mooring system, and stabilization approaches. As discussed in (Gottschall et al. (2017)), FLS can be categorized into buoy-based, spar-buoy, and semi-submersible platforms, each with distinct dynamic responses to wave-induced motion. These differences influence the magnitude of TI overestimation and should be carefully considered when interpreting FLS measurements. Furthermore, different lidar types and configurations, yield distinct estimates





of TI due to their varying spatial and temporal resolutions (Newman et al. (2016b)). The same applies to comparisons of lidar and cup anemometer derived TI (Sathe and Mann (2013); Newman et al. (2016a); Thiébaut et al. (2022)).

To eliminate the effects of motion on floating wind lidar measurements, several motion compensation approaches have been developed and introduced. Deterministic methods based on physical models are widely used in FLS TI motion compensation (Yamaguchi and Ishihara (2016); Kelberlau et al. (2020)). These methods rely on high-resolution motion data to directly

correct the orientation and measured radial velocities of the lidar's Line-of-Sight (LoS) measurements. The results are reliable and transparent when the platform's motion is precisely measured. However, they may struggle when faced with complex, non-linear motions, while relying on very accurate motion data, precise time stamping, and time synchronization.

Recent developments have explored data-driven methods utilizing machine learning (ML) (Rapisardi et al. (2024)). ML models are capable of learning complex, non-linear relationships between measurement errors and additional parameters e.g.

motion parameters or meteorological parameters, a feature that is not considered by traditional deterministic models. However, ML approaches require extensive, high-quality training data for the sea-states the FLS is experiencing and the regions its operating in. Theses models often operate as "black boxes", reducing transparency which may hinder their consideration and acceptance by standards.

Spectra-based models analyze the frequency content of the measured signals to differentiate genuine turbulence from motion-

induced noise, isolating the true turbulence signal by filtering out frequency bands dominated by motion artifacts (Thiébaut et al. (2024)). Numerical models simulate the dynamic interactions between wave-induced motions and lidar measurements, computationally quantifying and correcting biases in TI. These models integrate wave parameters, lidar scanning geometries, and platform responses to differentiate true atmospheric turbulence from motion-induced errors (Désert et al. (2021)). Statistical models, such as unscented Kalman filters estimate and correct TI biases by modeling uncertainties in wind measurements

and platform motion (Salcedo-Bosch et al. (2022). Hardware-based solutions, such as gimbals or other stabilization systems, provide a physical reduction of the motion on the measurements (Barros Nassif et al. (2020)). These methods increase system complexity and cost, but offer a direct way to compensate for motion in calmer sea states.

In this work, a deterministic approach described in Wolken-Möhlmann et al. (2010) is adopted, mainly because of the transparency, robustness as well as versatility of a physics-based correction model. By focusing on a deterministic motion

compensation, we ensure that even in scenarios with high motion dynamics, a consistent and understandable correction is achieved, presenting an advantage over less transparent, data-driven methods. Moreover, since the deterministic approach does not rely on training data, it can be deployed universally without being limited to specific conditions or locations.

The deterministic motion compensation is applied to FLS wind data from two FLS of the same type, equipped with different lidar types. The motion-compensated FLS TI data are then compared with raw FLS TI data, TI derived from the met mast cup

anemometer, and TI measurements from a fixed lidar. The objective is to eliminate motion-induced effects from the floating lidars TI measurements so that the compensated data closely resemble those from the fixed lidar. Using data from a fixed lidar of the same type as a baseline reference helps to isolate sensor-specific effects and validate the compensation performance.

To evaluate the effectiveness of the compensation, several metrics are analyzed. In (St. Pé et al. (2021)), the Consortium for Advanced Remote Sensing (CFARS), proposed to assess the accuracy of lidar-derived TI as a function of binned wind speed





using three key metrics: the TI Mean Bias Error (TI MBE), the TI Root Mean Square Error (TI RMSE), and the Representative TI Error (Representative TI). A similar formula to the Representative TI is mentioned in NEDO (New Energy and Industrial Technology Development Organization) (2023). Further research by Kelberlau et al. (2023) proposes acceptance thresholds based on their measurement data for TI MBE with a best practice threshold of 1.0% and a minimum practice threshold of 2.0% (absolute values) and for Representative TI with a best practice threshold of 1.5%, and a minimum practice threshold of 3.0% (absolute values), when compared to cup anemometer TI. Additionally, they propose applying Deming regression as an alternative to traditional ordinary least-squares (OLS) regression. While OLS assumes that all measurement errors are confined to the dependent variable, Deming regression accounts for uncertainties in both the independent and dependent variables. Furthermore, (DNV (2023)) introduces the Mean Relative Bias Error (TI MRBE) and the Relative Root Mean Square Error (TI RRMSE) to quantify the relative errors between the lidar-derived TI and cup anemometer measurements along with KPI thresholds for different use cases. The OWA Roadmap (OWA (2018)) recommends performing a regression through the origin (RTO) and calculating the slope and the coefficient of determination ($R^2$) to evaluate FLS TI against a trusted reference TI. However, the roadmap document does not define specific KPIs for TI accuracy. In (Uchiyama et al. (2024)), the TI measurements of several FLS are compared by performing regression analysis on the wind speed standard deviation, with a focus on bias. Furthermore, the 90th percentile of TI (Q90) as a function of the binned wind speed is assessed. Q90 is a key parameter used in wind turbine design and loads assessment (IEC (2019)). In IEC (2024), a Quantile-Quantile (Q-Q) analysis should be performed as part of the wind speed uncertainty assessment. In this work, this approach will be adapted for comparing TI measurement from different sources.

The paper is structured as follows: The methodology section (2) details the measurement equipment utilized (2.1), the applied motion compensation algorithm (2.2), and the specifications of the measurement campaign (2.4). It also presents the metrics used to assess the performance of the applied motion compensation (2.3). The results section (3) evaluates the TI measurements from the different FLS, both raw and motion-compensated, against reference data from a fixed lidar and a meteorological mast using the performance metrics described in the methodology section. In the discussion section (4), the findings are discussed, and the effectiveness of the deterministic motion compensation algorithm and its impact on the accuracy and precision of TI measurements emphasized. Finally, the conclusion section (5) summarizes the key findings of the study and suggests future directions for research.





## 2 Methodology

In this section, we outline the methodology employed in this study, which is structured into the following subsections. The measurement equipment is described in 2.1, detailing the Fraunhofer IWES Wind Lidar Buoy, the two deployed vertical profiling wind lidar types and the offshore meteorological mast. Following this, 2.2 focuses on the deterministic FLS motion compen-

sation algorithm applied in this study. The performance assessment metrics used to evaluate the effectiveness of the motion compensation are discussed in 2.3. Lastly, 2.4 provides details about the measurement campaign, including the deployment specification, data analysis time frame and environmental conditions.

### 2.1 Measurement equipment

In this subsection, we will introduce the measurement equipment utilized in this study. First, in 2.1.1, we present the character-

istics of the deployed FLS, specifically the Fraunhofer IWES Wind Lidar Buoy. Following this, 2.1.2 presents the two different vertical profiling lidar types used in this study: the continuous-wave (cw) vertical profiling wind lidar and the pulsed vertical profiling wind lidar, highlighting their differences. Lastly, in 2.1.3, we describe the offshore meteorological mast, which serves as a reference for the lidar measurements.

#### 2.1.1 Floating Lidar System

The Fraunhofer IWES Wind Lidar Buoy is based on the LT81 (Leuchttonne 1981) navigation buoy, a classic German navigation buoy design that has been deployed since 1981 (see Figure 1). Navigation buoys are engineered to maintain the visibility of their signal lights (or beacons) above the waterline at all times to safely guide ships. Fraunhofer IWES has adapted and optimized this design for wind measurement applications, replacing the beacon with an encapsulated wind lidar system.

The design specifications are listed in the following Table 1:





**Table 1.** Technical specifications of the Fraunhofer IWES Wind Lidar Buoy

| Parameter | Value |
| --- | --- |
| Buoy type | Fraunhofer IWES Wind Lidar Buoy |
| Dimensions | overall height 9.2 m, diameter 2.55 m |
| Weight | Approx. 5.6 t |
| Operational water depth | minimum 15 m |
| Material | Steel hull (DIN1.0036); Anodized aluminium for lidar housing |
| Mooring | DIN 5683-II mooring chain, 4.2 t concrete sinker, crowfoot (Y-shaped) arrangement |
| Data communication | WiFi, GSM, Iridium SBD, Iridium Certus, Starlink |
| Primary power system | Autonomous renewable energy-based power system, PV panels and three micro-wind turbines |
| Secondary power system | Diesel generator (running on GTL-Diesel) |
| IMU type | 1x Coda Octopus F175/F280, 2x Trimble BX992-INS |
| OWA stage | Stage 3 (ZX Lidars ZX300M cw lidar), Stage 2 (Vaisala Windcube 2.1 pulsed lidar) |

Due to its design and mooring characteristics, the Fraunhofer IWES Wind Lidar Buoy moves slowly in response to sea motion, effectively dampening wave-induced motions. This results in gradual, periodic movements rather than rapid displacements, helping to reduce high-frequency motion influences in lidar measurements, while reducing the requirements on motion data precision and time synchronization. However, as the buoy is experiencing motion in all six-degrees of freedom (see Figure 1 (b))), periodic variations are introduced in the lidars LoS measurements. These motion effects must be carefully accounted
for through motion compensation algorithms, to ensure the accuracy of TI measurements.



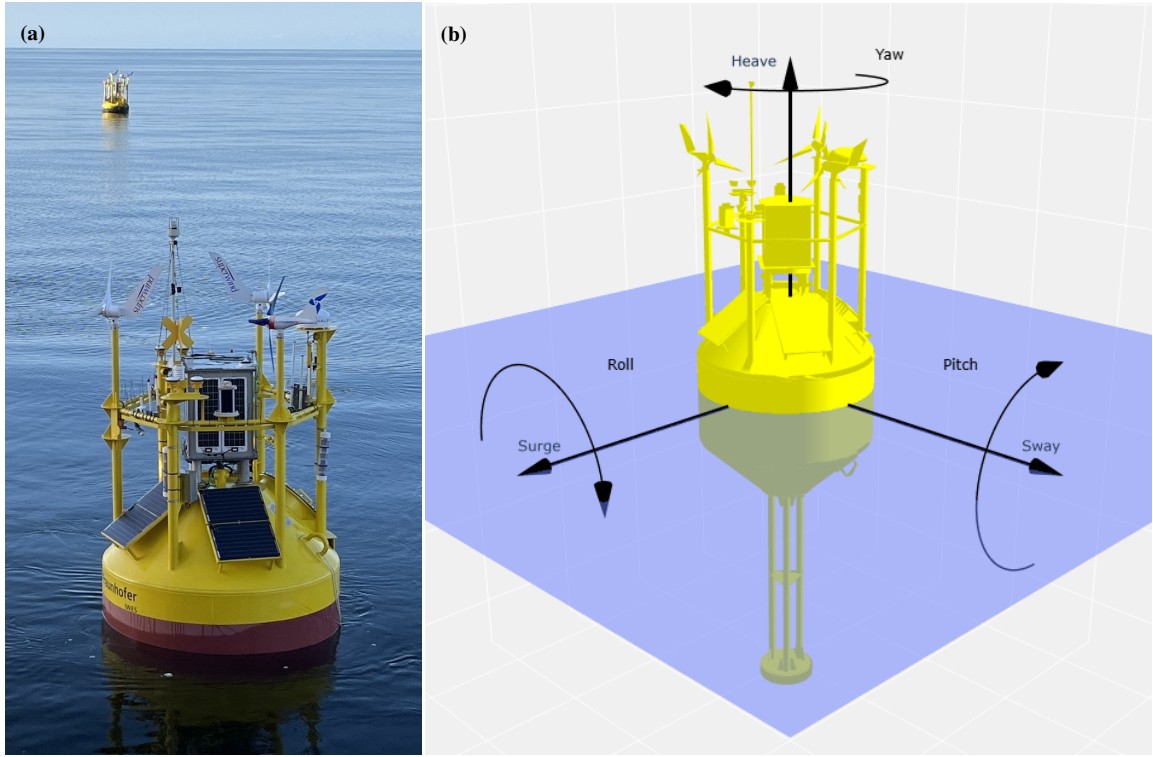

**Figure 1.** (a) Picture of the Fraunhofer IWES Wind Lidar Buoy (© Christian Tietjen, Fraunhofer IWES). In (b), a schematic representation of the Fraunhofer IWES Wind Lidar Buoy with a reference coordinate system and arrows denoting the 6-degrees of freedom is depicted.





### 2.1.2 Vertical profiling wind lidar technologies

In this study, two types of vertical profiling wind lidars were deployed and studied with respect to the the impact of motion on FLS TI measurements: the ZX Lidars ZX300M cw and the Vaisala Windcube V2.1 pulsed lidar system. The measurement principle of each lidar type is illustrated in the following Figure 2. The scan pattern of the cw lidar is depicted in (a), while (b) shows that of the pulsed lidar:

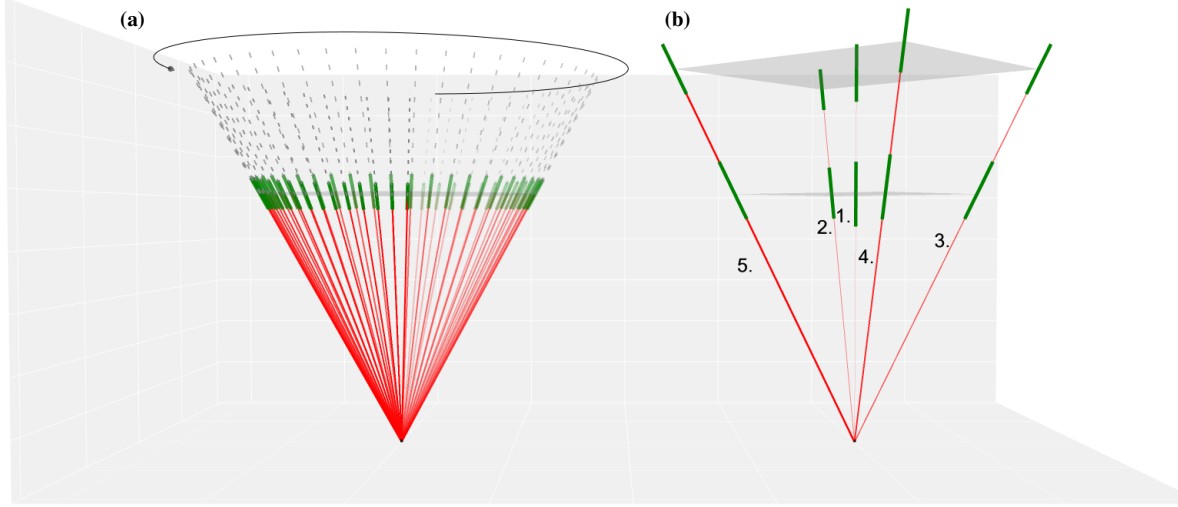

**Figure 2.** Simulation of vertical profiling wind lidar measurement principles. (a) illustrates the scanning geometry of a cw lidar performing a conical scan at 100 m; (b) depicts the scanning geometry of a pulsed lidar, measuring at two heights simultaneously, 100 m and 150 m.

The ZX300M cw vertical profiling lidar – (a) in Figure 2 – employs focusing optics to concentrate a continuously emitted light beam at predefined heights, conducting LoS measurements (depicted as red lines) in a conical scan pattern (sequence and direction implied by an increasing opacity of the red LoS lines and the gray arc arrow). Each scan cycle consists of 50 LoS measurements taken within a 1-second interval at one specific height. From each scan, a virtual wind vector is derived, using the velocity-azimuth-display (VAD) method before focusing on the subsequent measurement height. The predefined heights are scanned consecutively, and the total number of samples taken at each height within a 10-minute interval is depending on the number of specified measurement heights. The virtual wind vector is intended to represent the wind conditions over the lidar (gray area), ensuring that the resulting 10-minute average time series accurately reflects the wind speed and direction as measured by a meteorological mast (cup anemometers and wind vanes). Due to its focusing optics, the probe length of a cw lidar increases with measurement height. The green lines in 2 imply the probe length for a scan at 100 m ($\pm$ 7.70 m). The ZX300M is based on homodyne detection, which means that only the unsigned absolute value of the Doppler shift can be determined from the LoS measurements. Consequently, it cannot distinguish whether the wind is approaching or moving away, leading to a 180° ambiguity. To mitigate this ambiguity, the system relies on reference wind direction data supplied by an additional met station device typically installed closely above the lidar.





The Vaisala Windcube V2.1 pulsed lidar system – (b) in 2 – sequentially emits light pulses in four equally spaced 90° azimuth beam directions alongside one vertical beam (red lines in 2). The order of the sequence is marked with numbers as well as decreasing opacity of the red LoS lines. This scanning pattern is similar to a VAD and is often referred to as Doppler-beam-swinging (DBS). The time elapsed from the emission of each pulse is utilized to calculate the distance traveled by the light, thereby determining the measurement heights, referred to as range gates. The backscattered signal within each range gate is collected, and the LoS velocity is derived. From these LoS measurements, virtual wind vectors are calculated using the DBS method to represent the wind conditions at the corresponding height above the lidar (gray areas). The probe lenght of a pulsed lidar stays constant along all measurement heights. The green lines imply the probe length for a scan at $100\,\mathrm{m}$ and $150\,\mathrm{m}$ ($\pm$ $26.25\,\mathrm{m}$). The Windcube V2.1 deployed in this study was configured to collect LoS data at an increased frequency of $5\,\mathrm{Hz}$ with respect to the so far standard product with $1\,\mathrm{Hz}$.

### 2.1.3 FINO3 offshore met mast

The FINO3 (Forschungsplattform in Nord- und Ostsee 3) met mast is built on a monopile foundation that supports a platform and mast structure. The platform is located west of the DanTysk offshore wind farm, as shown in Figure 4. The mast cross section varies with height, which can result in stronger mast blockage effects at lower measurement altitudes (FINO3 Research Platform (2024)). FINO3 provides reference wind speed measurements at multiple heights using cup and sonic anemometers. For this study, only cup anemometers that are mounted on booms with the same orientation (345°N) are selected, allowing the use of the same free-inflow sector for all altitudes. Wind vanes for measuring wind direction are located at heights of $101\,\mathrm{m}$ and $29\,\mathrm{m}$.

### 2.2 Floating lidar motion compensation

The deterministic motion compensation algorithm considers the tilted and rotated LoS beam vectors, as well as the velocities induced by surge, sway, heave and tilt motions to account for the motion-induced change in the measured radial velocity as described in (Wolken-Möhlmann et al. (2010)). Following the same principle as in Figure 2, Figure 3 visualizes extreme events of tilted LoS beam vectors of a floating cw lidar (a) and floating pulsed lidar (b) from actual measurement data.



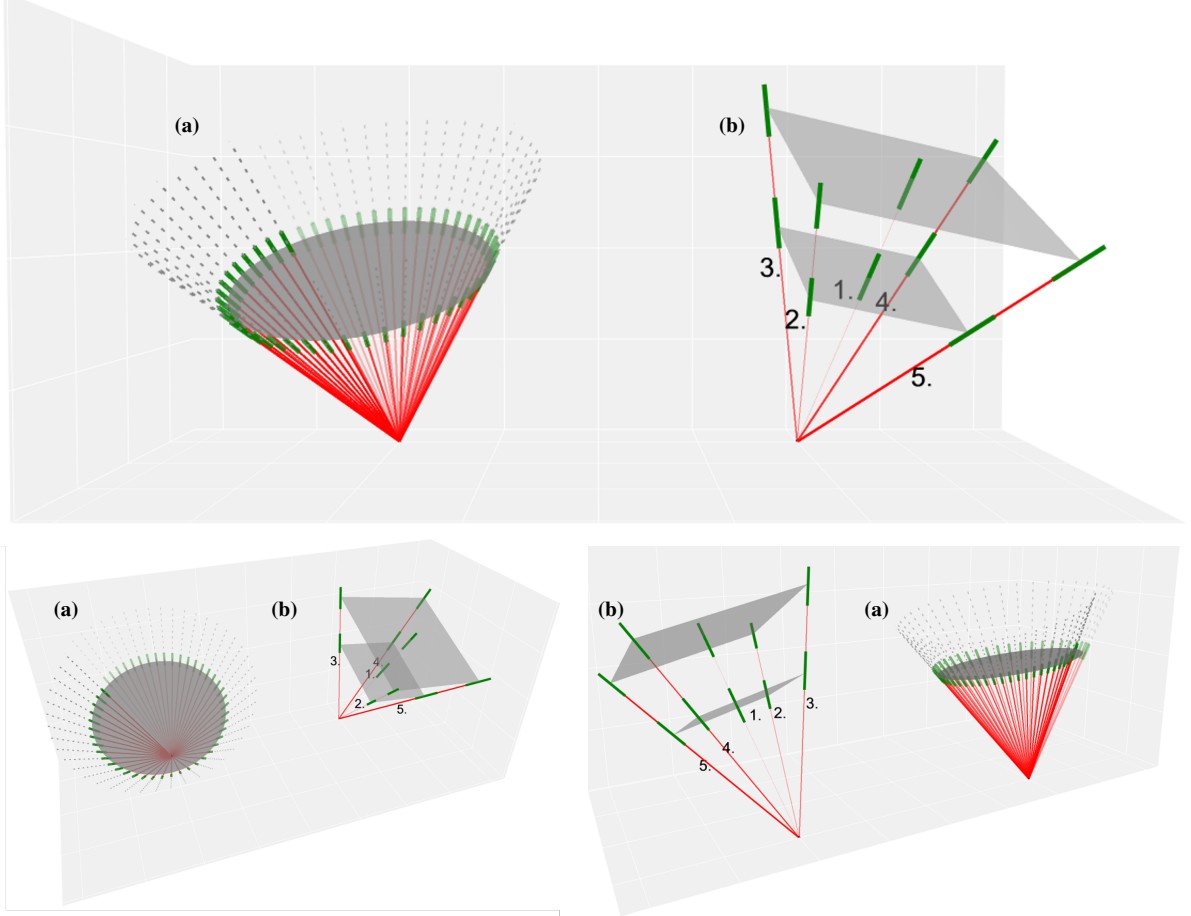

**Figure 3.** Simulation of tilted vertical profiling wind lidar measurement principles. (a) illustrates the scanning geometry of a cw lidar performing a conical scan at 100 m; (b) depicts the scanning geometry of a pulsed lidar, measuring at two heights simultaneously, 100 m and 150 m.

185    As shown in 3, the tilting, yawing and heave directly influence the scanning geometry of the lidar systems, deforming the scan volume. The changed measurement height (due to tilt and heave) is considered in different ways, depending on the lidar type. As pulsed lidar systems take measurements almost simultaneously (see 2.1.2) at multiple heights, the LoS velocity of each beam can be interpolated to the desired measurement height by considering adjacent measurements from neighboring heights of the same pulse. Continuous-wave lidars conduct sequential measurements, focusing on one measurement height before

190    progressing to the next. Consequently, the interpolation method is not applicable. To account for changes in measurement height, the power law wind profile (IEC (2019)) is applied to the radial wind velocity of each transformed LoS. After applying the transformation matrices to the LoS beams and compensating the measured radial velocities for the motion as described, the virtual wind vector is calculated by applying the VAD or DBS method depending on the instrument.





Potential time offsets between the lidar and the motion sensor data are identified by repeating the motion compensation several times for each 10-minute interval, while shifting the lidar data by small time increments (time offsets). The standard deviation of each resulting 10-minute wind interval is determined, and the interval with the lowest standard deviation is selected. A lower standard deviation indicates less fluctuation and, consequently, a reduced influence of motion on the measured wind data. The corresponding time offset then becomes the basis for the offset iteration of the next 10-minute interval. Depending on the lidar type, the time offset increases over time until the device resynchronizes, at which point the time offset returns to zero. To consider this resynchronization, the time-offset iteration always includes a time offset of zero.

## 2.3 Performance assessment metrics

In this subsection, we introduce the performance assessment metrics used to evaluate the TI measurements. Building on the wind industry's standard practice of analyzing 10-minute statistical intervals, we derive time series data that can be statistically analyzed, for example, through linear regression. For standard parameters, such as mean wind speed and direction, this approach yields the slope, offset, and $R^2$, for which KPIs are defined in (OWA (2018)). A similar methodology can be applied to TI, while alternative metrics might be more directly connected to the application of TI.

We begin by describing the application of the linear regression techniques in 2.3.1 to capture the overall trend between the TI measurements. 2.3.2 then introduces the Mean Bias Error and Mean Relative Bias Error, which are used to quantify systematic differences between the datasets. In 2.3.3, we describe the calculation of the Root Mean Square Error and its normalized variant (Relative Root Mean Square Error), which highlight the magnitude of random error or precision. 2.3.4 focuses on the Representative TI Error, derived from the 90th percentile of the TI distribution. Finally, 2.3.5 presents a quantile-based distribution analysis, that provides a visual assessment of measurement distributions and potential biases.

### 2.3.1 Linear regression

Linear regression and correlation analysis are commonly used to compare wind speed and direction measurements from multiple sources, such as in calibration campaigns or for plausibility checks of data. While OLS assumes that all measurement errors are confined to the dependent variable, Deming regression accounts for uncertainties in both the independent and dependent variables. In this study, we will investigate the performance using OLS, RTO and Deming regression for comparison. While the uncentered $R^2$ is recommended for RTO, it has been observed to produce abnormally high values, which distort the results. The uncentered $R^2$ ($R^2_{\text{uncentered}}$) is described as follows:

$$R^2_{\text{uncentered}} = 1 - \frac{\sum_{n=1}^{N} \left( \text{TI}_{\text{ref},n} - \text{TI}_{\text{comp},n} \right)^2}{\sum_{n=1}^{N} \left( \text{TI}_{\text{ref},n} \right)^2} \tag{1}$$

where $\text{TI}_{\text{comp}}$ is the comparison quantity, $\text{TI}_{\text{ref}}$ is the reference quantity, $n$ refers to the individual data point, and $N$ is the total number of data points.





To calculate $R^2$, we will use the following equation as an alternative to the uncentered $R^2$:

$$R^2 = 1 - \frac{\sum_{n=1}^{N} \left( \mathrm{TI}_{\mathrm{ref},n} - \mathrm{TI}_{\mathrm{comp},n} \right)^2}{\sum_{n=1}^{N} \left( \mathrm{TI}_{\mathrm{ref},n} - \overline{\mathrm{TI}}_{\mathrm{ref}} \right)^2} \tag{2}$$

Using this equation, $R^2$ might yield negative values when the predictions are worse than simply using the mean of the observed data as a predictor. In that case, we will print Not a Number (NaN).

### 2.3.2  Mean bias error and mean relative bias error

The TI MBE as a function of binned wind speed is defined as follows (St. Pé et al. (2021)):

$$\mathrm{TI}_{\mathrm{fig:ti_mbe},i} = \frac{1}{N_i} \sum_{n=1}^{N_i} \mathrm{TI}_{\mathrm{comp},n,i} - \mathrm{TI}_{\mathrm{ref},n,i} \tag{3}$$

where $i$ is the wind speed bin (with a bin size of $1\,ms^{-}1$), and $N_i$ is the number of data points in the $i$th bin. The TI MBE measures the average difference between two datasets, helping to identify any consistent deviations. It reveals systematic over- and underestimations (bias), thus indicating the direction of the error.

The TI MRBE, as introduced by DNV (2023), is deployed to derive the relative error (relative TI bias) between lidar and cup anemometer TI. It is a variation of the Mean Bias Error that expresses bias as a normalized measure. It is defined as follows:

$$\mathrm{TI}_{\mathrm{MRBE},i} = \frac{1}{N_i} \sum_{n=1}^{N_i} \frac{\left( \mathrm{TI}_{\mathrm{comp},n,i} - \mathrm{TI}_{\mathrm{ref},n,i} \right)}{\mathrm{TI}_{\mathrm{ref},n,i}} \times 100 \tag{4}$$

The metric again reveals systematic over- and underestimations but by normalizing the bias, the error becomes easier to interpret.

### 2.3.3  Root mean square error and relative root mean square error

The TI RMSE as a function of binned wind speed is denoted as follows (St. Pé et al. (2021)):

$$\mathrm{TI}_{\mathrm{RMSE},i} = \sqrt{\frac{1}{N_i} \sum_{n=1}^{N_i} (\mathrm{TI}_{\mathrm{comp},n,i} - \mathrm{TI}_{\mathrm{ref},n,i})^2} \tag{5}$$

The TI RMSE quantifies the magnitude of random errors between two datasets by calculating the square root of the average squared differences. By focusing on squared differences, it highlights the random errors and statistical variability that may arise due to differences in measurement instruments.

Normalizing the TI RMSE yields the TI RRMSE that enables direct comparisons across different datasets. It is defined as 245  (DNV (2023):

$$\mathrm{TI}_{\mathrm{RRMSE},i} = \sqrt{\frac{1}{N_i} \sum_{n=1}^{N_i} \left( \frac{\left( \mathrm{TI}_{\mathrm{comp},n,i} - \mathrm{TI}_{\mathrm{ref},n,i} \right)}{\mathrm{TI}_{\mathrm{ref},n,i}} \right)^2} \times 100 \tag{6}$$





### 2.3.4 Representative TI error

The Representative TI is defined as the 90th quantile (Q90) of a TI dataset. For Gaussian distributions, the Q90 can be approximated according to (St. Pé et al. (2021)):

$$\text{TI}_{\text{Rep},i} = \text{TI}_{\text{avg},i} + 1.28 \times \text{TI}_{\text{std},i} \tag{7}$$


where $\text{TI}_{\text{avg}}$ is the average TI, and $\text{TI}_{\text{std}}$ is the standard deviation of the TI.

We also apply this equation in our study, instead of calculating the Q90 based on the data population per bin. The Representative TI error is the difference between the binned Representative TI values of the reference and trialed system:

$$\text{TI}_{\text{Rep}_{\text{error}},i} = \frac{1}{N_i} \sum_{n=1}^{N_i} (\text{TI}_{\text{Rep,comp},i} - \text{TI}_{\text{Rep,ref},i}) \tag{8}$$

### 2.3.5 Quantile-based distribution analysis

A Q-Q analysis is a graphical tool used to compare the distributions of datasets by plotting their quantiles against each other. It helps assess the goodness of fit between datasets, allowing for the visual identification of whether the data follows the reference distribution or shows patterns of systematic error or outliers. Additionally, overestimation and underestimation (systematic bias) can be derived from the plot by examining the position of the data points relative to the 1:1 line.





## 2.4 Campaign details

Two FLS of the type Fraunhofer IWES Wind Lidar Buoy were deployed in proximity to the FINO3 offshore met mast from March 6, 2024, to August 3, 2024. The distance between the FLS and the met mast was approximately 300 m. One FLS was equipped with a ZX Lidars ZX300M cw lidar system (FLS ZX), the other one with a Vaisala Windcube V2.1 pulsed lidar in a 5 Hz scanning configuration (FLS WC). Additionally, a ZX Lidars ZX300M cw lidar system (Fixed ZX) was installed on the FINO3 met mast platform in a height of approx. 27 m above LAT. Both cw lidar systems were running the same firmware version (v3.3002).

An overview of the device characteristics and configurations is given in Table 2. Matching wind speed measurement heights are marked in bold:

| | Met Mast | Fixed ZX | FLS ZX | FLS WC |
|---|---|---|---|---|
| **Measurement device type** | Cup Anemometer | CW lidar | CW lidar | Pulsed lidar |
| **Model** | A100L2 | ZX300M | ZX300M | Windcube V2.1 |
| **Manufacturer** | Windspeed LTD | ZX Lidars | ZX Lidars | Leosphere (Vaisala France) |
| **LoS sampling rate [Hz]** | Not applicable | 50 | 50 | 5 |
| **Half-cone opening angle [°]** | Not applicable | 30.6 | 30.6 | 28 |
| **Probe length [m]** | Not applicable | ± 0.07 at 10 m; ± 7.70 at 100 m** | ± 0.07 at 10 m; ± 7.70 at 100 m** | ±26.25 (constant)*** |
| **Affected by motion** | No | No | Yes | Yes |
| **Measurement heights [m]** | 31, 41, 51, 61, **71**, 81, **91**, **101**, **107*** | 64, **71**, **91**, **101**, **107**, 130, 160, 200, 225, 250, 275 | 42, **71**, **91**, **101**, **107**, 130, 160, 200, 225, 250 | **71**, **91**, **101**, **107**, 130, 160, 200, 225, 250, 275 |

**Table 2.** Measurement device characteristics.

* (FINO3 Research Platform (2024))

** (ZX Lidars (2024))

*** Calculated according to Equation 29 in (Pena (2009)).

The positions of all systems are shown in the following Figure 4:





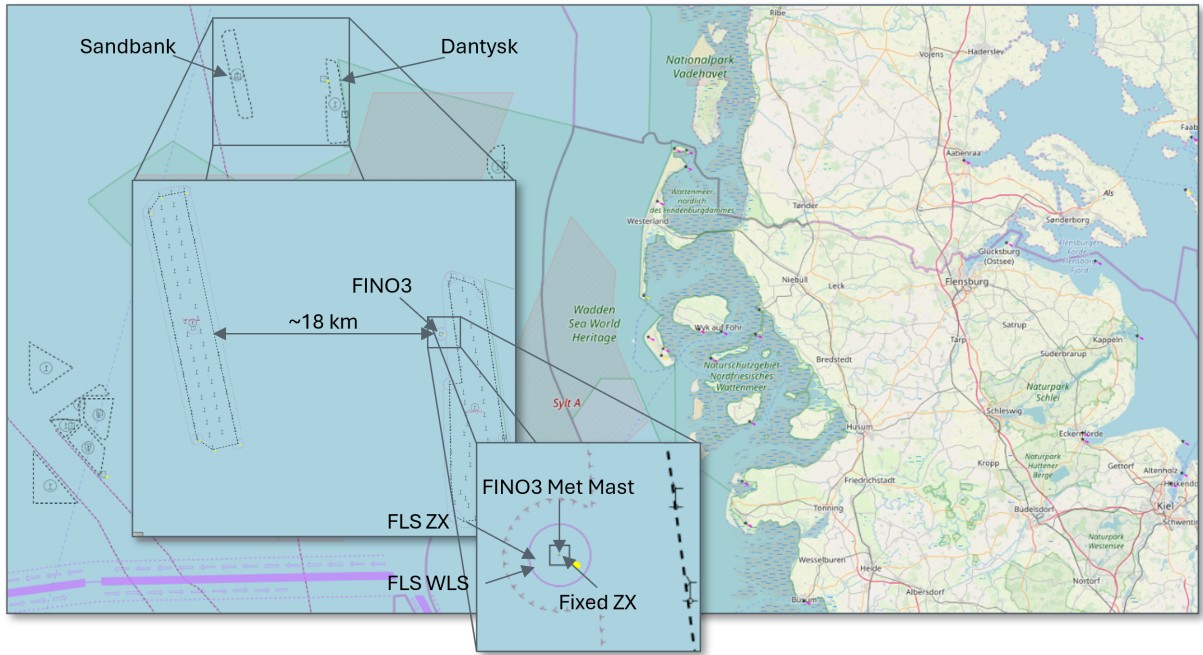

**Figure 4.** Measurement campaign location at FINO3 in the German North Sea (based on OpenSeaMap Data (OpenSeaMap (2025)) under the Open Database License (ODbL)).

To enhance the accuracy of the motion compensation results, the settings of the motion sensors deployed on the FLS were optimized on April 6, 2024. The fixed cw lidar installed on the FINO3 met mast platform (Fixed ZX) stopped recording data due to a power supply failure on July 9, 2024, as a result only data recorded from April 6 to July 9 was considered in this study. Additionally, only timestamps where data from all measurement systems were available were considered. A wind sector filter is applied to only consider wind data coming from the free inflow sector 220° to 300°, using the FINO3 met mast wind vane

installed on 101 m above LAT as reference.

The met mast and wave radac reference datasets were obtained from the BSH Insitu database (für Seeschifffahrt und Hydrographie (BSH)). All entries flagged as questionable or bad were removed from the analysis. The measurement campaign covered a range of environmental conditions with average wind speeds reaching up to $26.91\,ms^-1$ (measured by the cup at 107 m), significant wave heights (Hs) reaching up to 4.28 m and peak wave periods (Tp) of up to 14.3 s. The time series of the wind

and sea-state conditions recorded during the trial period are visualized in Figure 5.



**Figure 5.** Met mast cup wind speed (a) and vane wind direction with the sector filter marked as dashed lines (b), as well as radac significant wave height (Hs) (c) and (d) peak period (Tp) time series during the trial period from 2024-04-06 00:00:00 to 2024-07-08 23:50:00.

No data was recorded by the cup anemometer mounted on 91 m. Furthermore, a data gap is present in the radac Hs and Tp records from 2024-06-07 09:00:00 to 2024-06-19 07:50:00 UTC, which may affect the statistics of certain sea state parameters which are shown in Figure 6. Panel (a) presents the wind speed distribution, measured by the cup anemometer at 101 m LAT. Panel (b) displays a wind rose, illustrating the distribution of wind directions recorded by the wind vane at 101 m LAT, along with the corresponding bin-wise wind speed distribution measured by the cup anemometer at the same height. Panel (c) shows





the normalized frequency distribution of the significant wave height, measured by the radac mounted on the met mast. Lastly, panel (d) presents a density correlation plot of significant wave height versus spectral peak period measured by the radac mounted on the met mast.



**Figure 6.** Statistical overview of reference conditions recorded during the trial period: (a) wind speed distribution measured by the cup anemometer at 101 m LAT; (b) wind rose displaying wind direction distribution measured by the wind vane at 101 m LAT along with the corresponding bin-wise wind speed distribution from the cup anemometer at the same height; (c) normalized frequency distribution of the significant wave height measured by the radac mounted on the met mast; (d) significant wave height versus spectral peak period density correlation plot measured by the radac mounted on the met mast.





# 3  Results

In this section, the TI data recorded during the measurement campaign is evaluated. The analysis focuses on TI measurements from different lidar types (both raw and motion-compensated) installed on two FLS and their comparison with reference data from a fixed cw lidar and cup anemometers mounted on a meteorological mast. The aim has been to analyze the effect of the deterministic motion compensation on the FLS TI data.

The results are structured as follows: First in 3.1, a correlation analysis is conducted to compare the TI measured by the
FLS before and after motion compensation. This is followed by an assessment of systematic deviations through MBE and MRBE in 3.2. After that in 3.3, precision metrics, namely RMSE and RRMSE are evaluated. Additionally, the Representative TI error is examined in 3.4. Finally in 3.5, a quantile-based distribution analysis is conducted to provide further insights into the distribution of FLS TI before and after motion compensation.

To ensure consistency in the analysis, a wind sector filter for the range of 220° to 300° was applied, based on wind direction
data from the met mast wind vane at 101 m above LAT. Additionally, for the regression analysis and the Quantile-based distribution analysis, only wind speed data within the range of $4\,ms^{-}1$ to $16\,ms^{-}1$, as measured by the met mast cup anemometer at the same height, were considered. The results presented in this section are based on measurements from 101 m above LAT. The results of further measurement heights (71 m, and 107 m) are found in Appendix A.





### 3.1 Linear regression and correlation analysis

For this study, the fixed cw lidar (Fixed ZX) TI serves as the baseline for a lidar-derived TI measurement without the influence of motion. This baseline sets the benchmark for assessing the performance of the applied motion compensation algorithm. Figure 7 presents a correlation plot comparing fixed cw lidar TI with met mast cup anemometer (MM) TI at 101 m above LAT. To highlight the distribution of data points, the scatter is represented as a density plot, where point density is indicated by a color scale. A solid black 1:1 line represents perfect agreement between the two measurements. As mentioned in 2.3.1, three

regression models are analyzed, with $R^2$ calculated according to Equation 2: OLS regression (red dashed line), RTO (red solid line), and Deming regression (black dashed line). The regression parameters (slope, offset, and $R^2$ values) are listed in Table 3 for reference.

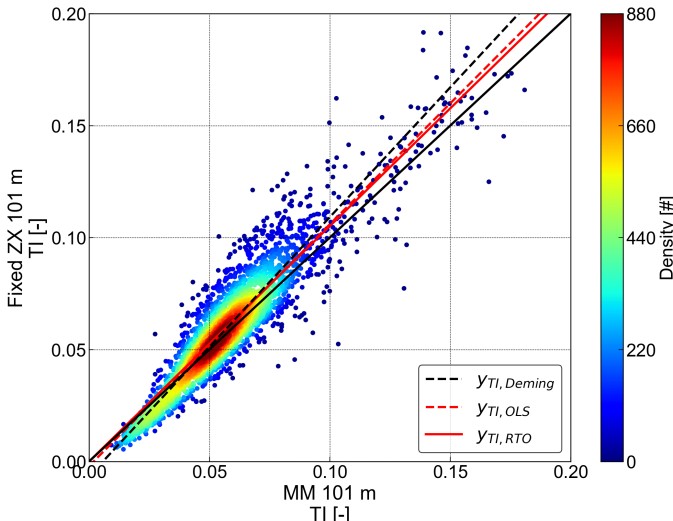

**Figure 7.** Density correlation plot and regression analysis of the fixed cw lidar (Fixed ZX) TI versus the met mast cup anemometer (MM) TI at 101 m above LAT. The point density is indicated by the color bar. Derived parameters are listed in Table 3.

**Table 3.** Regression parameters for the correlation between TI measured by the fixed cw lidar (Fixed ZX) and the met mast cup anemometer (MM) at 101 m above LAT, as illustrated in Figure 7.

|  | Deming | OLS | RTO |
|---|---|---|---|
| **Slope** | 1.158 | 1.080 | 1.051 |
| **Intercept** | -0.007 | -0.002 | - |
| **$R^2$** | 0.880 | 0.884 | 0.884 |
| **Number of 10-minute data points** | | 3,034 | |

The data points are well aligned along the 1:1 line, indicating a strong correlation between the fixed cw lidar and the met mast cup measurements. All three regression models yield $R^2$ values above 0.8. While wind speed and wind direction correlations





typically exhibit even higher $R^2$ values, this is considered high for a TI correlation. However, a slight overestimation of TI by the fixed cw lidar can be observed, particularly at higher values. The applied regression models confirm this trend with consistent slopes above unity and only minor offsets. Further, the resulting Deming regression slope is higher than the slopes from OLS and RTO, indicating that the cw lidar measurements systematically show higher TI values than the met mast cup reference. As mentioned in Section 2.3, the outcome of a regression analysis for TI are significantly influenced by several

factors. Although the fixed cw lidar is installed on the met mast platform, the visible scatter as well as the slope and $R^2$ of the three types of regression fits between the two instruments, reveal deviations between the datasets. These discrepancies are primarily caused by the differing underlying measurement principles of the instruments.

Building on the baseline analysis in Figure 7, we now compare motion-affected cw lidar data from the FLS (FLS ZX) at 101 m above LAT to the fixed cw lidar TI at the same height. Figure 8 presents two scatter plots, following the same approach as in

Figure 7 with (a) showing the raw (uncorrected) FLS measurements, while (b) displays the motion-compensated results. The corresponding regression parameters are listed in Table 4. As the floating lidar is subject to wave-induced motion, additional velocity fluctuations are introduced, leading to an overestimation of TI compared to the fixed cw lidar. By comparing raw and motion-compensated FLS data, we assess the extent of motion-induced bias and evaluate the effectiveness of the applied correction algorithm in mitigating these effects.

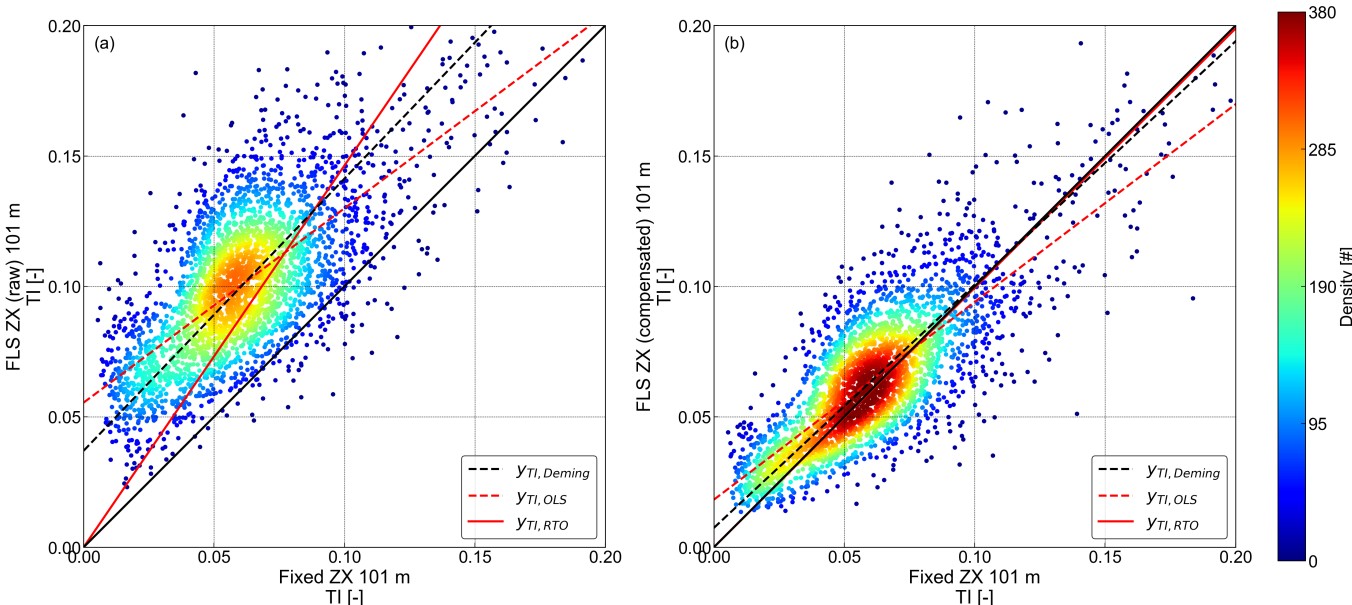

**Figure 8.** Density correlation plots and regression analyses of TI measured by the floating cw lidar (FLS ZX) versus the fixed cw lidar (Fixed ZX) at 101 m above LAT. Panel (a) shows uncompensated (raw) data, while panel (b) shows motion-compensated data. Point density in both panels is indicated by the color bar. Derived parameters are listed in Table 4





**Table 4.** Regression parameters for the correlation between TI measured by the floating cw lidar (FLS ZX) and the fixed cw lidar (Fixed ZX) at 101 m above LAT, as illustrated in Figure 8.

|  | Panel (a): Raw data | | | Panel (b): Motion-compensated data | | |
|---|---|---|---|---|---|---|
|  | **Deming** | **OLS** | **RTO** | **Deming** | **OLS** | **RTO** |
| **Slope** | 1.043 | 0.745 | 1.463 | 0.933 | 0.758 | 0.994 |
| **Intercept** | 0.037 | 0.055 | - | 0.007 | 0.018 | - |
| $R^2$ | 0.438 | 0.522 | NaN | 0.608 | 0.642 | 0.564 |
| **Number of 10-minute data points** | 3,034 | | | 3,034 | | |

Figure 8 visualizes the difference between the raw (a) and motion-compensated (b) FLS TI measurements.

In (a), a noticeable scatter and deviation from the 1:1 line indicate that the floating lidar systematically overestimates TI. This is reflected in the regression parameters listed in Table 4, where all models show lower $R^2$ values compared to the motion-compensated variant in (b). In the raw cw FLS data, the Deming regression results in a slope of 1.043 and an offset of 0.037, with an $R^2$ of 0.438, indicating a moderate correlation alongside a systematic overestimation reflected in the high offset. The

OLS regression produces a much lower slope of 0.745 alongside an offset of 0.055, further suggesting that the raw cw FLS TI measurements tend to diverge significantly from the fixed cw lidar TI. The RTO regression, with a slope of 1.463, suggests a high overestimation, while its negative $R^2$ value underlines the poor fit (and is therefore marked as NaN).

After motion compensation for (b), the scatter is reduced, and the correlation between cw FLS and fixed cw lidar significantly improves across the three analyzed regression models. The Deming regression now results in a slope below one and an minor

offset of 0.007, with an increased $R^2$ of 0.608, suggesting that the overestimation seen in the raw cw FLS data has largely been corrected, while a slight underestimation trend was introduced. The OLS regression shows a similar slope as in the raw cw FLS data, but with a much lower offset and an increased $R^2$, indicating that the motion compensation has not only reduced the bias but also reduced the scatter between the two datasets. The RTO regression, with a slope of 0.994 and an $R^2$ of 0.564, now provides a reasonable fit, supporting the improvement of the data quality.

These results confirm that the applied motion compensation algorithm effectively mitigates the motion-induced overestimation seen in the raw cw FLS TI data. However, the slopes below one in the compensated dataset suggest a slight underestimation, particularly in the Deming regression. Several factors contribute to these remaining discrepancies. The deterministic motion compensation algorithm relies on accurately resolving the signs of the measured LoS velocities to correctly adjust for motion effects. This process becomes particularly challenging in conditions of non-uniform flow at low wind speeds and high TI, pri-

marily due to the homodyne detection of the Doppler shift in cw lidars (see Section 2.1.2) and the assumption of homogeneous flow in the VAD scanning pattern. These limitations introduce errors in the derived virtual wind vectors, which can lead to additional fluctuations or undercompensation of motion-induced effects. Further factors are for example the remaining time-offsets between the IMU and lidar device, poor time stamping precision, the distance between the instruments (see Section 2.4), the different probe volume and the smaller focal length of the elevated reference lidar compared to the floating cw lidar.






While Figure 8 has focused on the comparison between the floating and fixed cw lidar TI, the next step has been to investigate whether similar trends are observed for the floating pulsed lidar system (FLS WC). Figure 9 extends the analysis by comparing raw and motion-compensated TI measurements from the floating pulsed lidar (FLS WC) against the fixed cw lidar reference at 101 m above LAT following the same approach as in Figure 7 and Figure 8. Since pulsed lidars differ in their measurement principles, particularly the scanning geometry and range gating (see Section 2.1.2), the impact of motion and the effectiveness of the compensation algorithm is expected to show different characteristics compared to the floating cw lidar system.

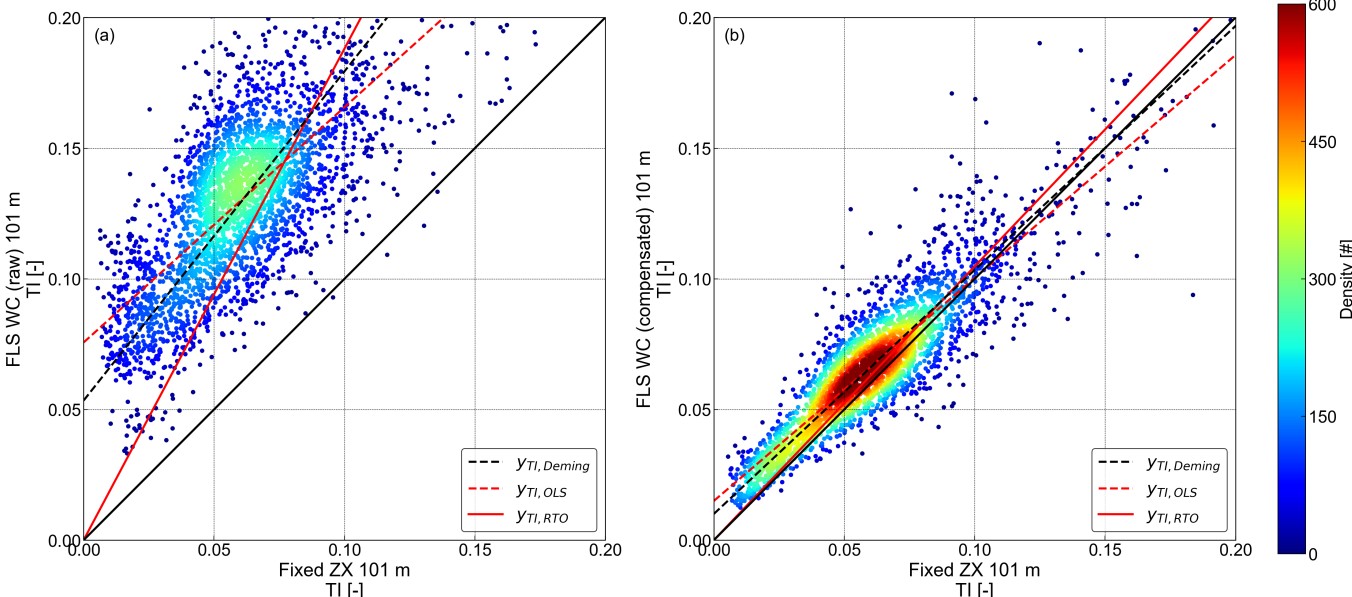

**Figure 9.** Density correlation plots and regression analyses of TI measured by the floating pulsed lidar (FLS WC) versus the fixed cw lidar (Fixed ZX) at 101 m above LAT. Panel (a) shows uncompensated (raw) data, while panel (b) shows motion-compensated data. Point density in both panels is indicated by the color bar. Derived parameters are listed in Table 5

**Table 5.** Regression parameters for the correlation between TI measured by the floating pulsed lidar (FLS WC) and the fixed cw lidar (Fixed ZX) at 101 m above LAT, as illustrated in Figure 9.

| | Panel (a): Raw data | | | Panel (b): Motion-compensated data | | |
|---|---|---|---|---|---|---|
| | **Deming** | **OLS** | **RTO** | **Deming** | **OLS** | **RTO** |
| **Slope** | 1.260 | 0.900 | 1.881 | 0.934 | 0.854 | 1.048 |
| **Intercept** | 0.053 | 0.076 | - | 0.010 | 0.015 | - |
| $R^2$ | 0.480 | 0.571 | NaN | 0.817 | 0.824 | 0.771 |
| **Number of 10-minute data points** | | 3,034 | | | 3,034 | |





In Figure 9, the scatter plots reveal distinct differences between the raw (a) and motion-compensated (b) TI measurements from the floating pulsed lidar (FLS WC). The raw dataset exhibits wide scatter, with data points deviating significantly from the 1:1 line. This increased dispersion, along with a consistent upward shift, suggests a systematic overestimation of TI and highlights the strong influence of platform motion on the floating pulsed lidar measurements.

The regression models further confirm this trend. Deming regression yields a slope of 1.260 with a high offset of 0.053, while OLS regression results in a slope of 0.900 and an even higher offset of 0.076, both reflecting the overestimation. RTO yields a slope of 1.881 with a negative $R^2$ underlining the severe overestimation but also revealing the poor fit.

In contrast, the motion-compensated dataset (b) exhibits a clear reduction in scatter and overestimation, with data points and regression fits aligning closely with the 1:1 line. This improvement is further reflected in significantly reduced offsets and increased $R^2$ values across all models and regression slopes generally shifting closer to unity. The density scatter plot reveals an overestimation of TI for lower values, tilting the derived slopes. The RTO slope decreases from 1.881 to 1.048, now closely aligning with the 1:1 line. Meanwhile, the Deming regression slope decreases from 1.260 to 0.934, and OLS from 0.900 to 0.854, indicating that the compensation effectively mitigates motion-induced variability but seems to introduces a slight underestimation. The overall increase in $R^2$ values confirms that the compensation algorithm successfully corrects the TI measurements and significantly improves agreement with the fixed cw lidar reference. The remaining discrepancies may be attributed to similar factors as previously mentioned, with the added influence of the different underlying measurement principles of the compared lidars.

## 3.2 Mean Bias Error and Mean Relative Bias Error

The figures in 2.3.1 demonstrated how the applied deterministic motion compensation reduced scatter and systematic overestimation in floating lidar TI measurements. While the regression analysis provided insights into the overall relationship between the datasets, a more detailed performance assessment is conducted by evaluating systematic deviations and measurement accuracy across different wind speed bins. To achieve this, the following analysis examines further performance assessment metrics, as introduced in Section 2.3.

The following Figure 10 illustrates the binned MBE (calculated according to Equation 3 in Section 2.3.2) between the met mast cup TI and the trialed lidar devices TI as a function of binned wind speed. The figure includes both raw and motion-compensated datasets (where applicable), distinguished by dashed and solid lines, respectively. The x-axis represents the wind speed bins, while the y-axis displays the corresponding error metric values. The figure also features minimum and best practice performance thresholds, indicated by dashed horizontal lines.





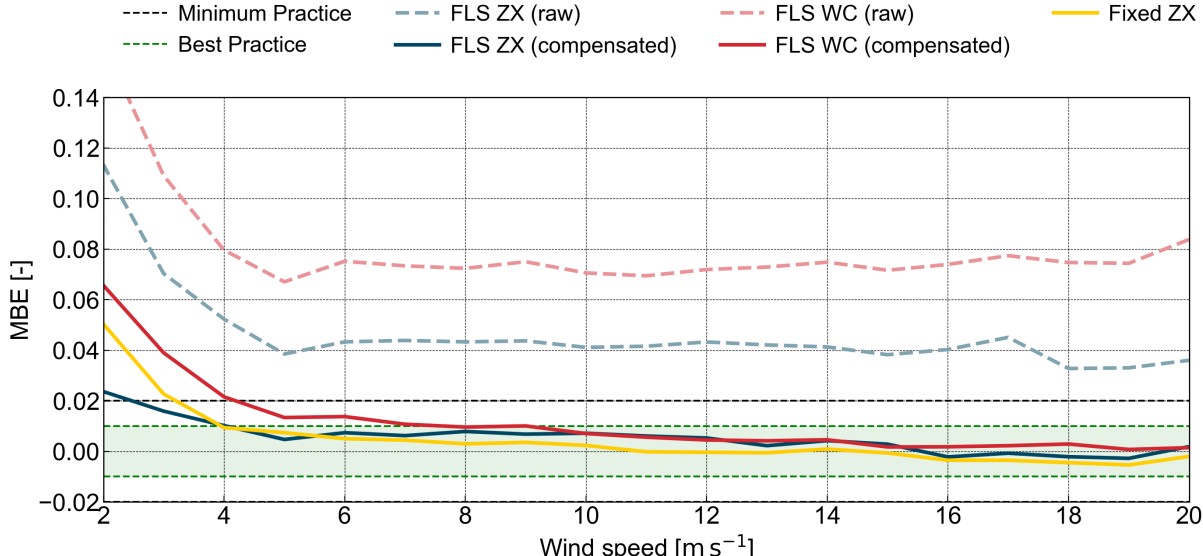

**Figure 10.** Binned Mean Bias Error between the MM TI and the TI of the trialed devices Fixed ZX (yellow), FLS ZX (blue lines, where the dashed line represents raw data and the solid line depicts motion-compensated data) and the FLS WC (red lines, where the dashed line represents raw data and the solid line depicts motion-compensated data) from 101 m above LAT.

The MBE trends in Figure 10 demonstrate a clear distinction between the raw and motion-compensated datasets, illustrating the systematic overestimation of TI in the uncompensated data and the effectiveness of the compensation algorithm. Both raw datasets, represented by the dashed red and blue lines, consistently exhibit a positive bias across all wind speed bins. This bias is most pronounced at lower wind speeds below $5\,ms^{-1}$, where motion-induced fluctuations have a greater relative impact on TI due to the increased influence of platform movement in relation to wind speed. Moreover, TI values are generally higher at lower wind speeds, increasing the potential for greater bias in this range.

While both lidar types exhibit systematic TI overestimation in their raw datasets, the raw floating pulsed lidar TI (red dashed line) consistently shows a higher positive bias than the raw floating cw lidar TI (blue dashed line), while following a similar pattern until diverging for wind speeds higher than $17.5\,ms^{-1}$. This suggests that the pulsed lidar is more sensitive to motion-induced fluctuations, likely due to its sequential scanning method. In contrast, the cw lidar, which averages LoS velocities over a conical scan, appears to be less affected by motion variations, resulting in a lower overall bias in the raw data.

The fixed cw lidar also shows a relatively high bias at low wind speeds, which steadily declines until it approaches near-zero bias between $5\,ms^{-1}$ and $16\,ms^{-1}$. While this trend highlights the inherent differences between TI measurements from cw lidars and those derived from a cup anemometer, mast effects might also influence the measurements. Following motion compensation, the bias in both the floating pulsed (red solid line) and cw lidar (blue solid line) datasets is significantly reduced, confirming the effectiveness of the applied compensation algorithm. The floating pulsed lidar exhibits the most noticeable relative improvement, with a steep decline in bias at low wind speeds and a further reduction as wind speed increases. Despite this, the bias remains consistently positive across all wind speed bins, indicating that while compensation effectively mitigates





motion effects, a small residual overestimation persists. For wind speeds between $4\,ms^{-}1$ and $9\,ms^{-}1$, the pulsed lidar falls well within the minimum practice range, before transitioning into the best practice area for all higher wind speeds. The floating cw lidar maintains a low bias across all wind speed bins. Between $2.5\,ms^{-}1$ and $4\,ms^{-}1$, it remains below the 0.02 MBE line, before entering the best practice area. At $5\,ms^{-}1$, its error is slightly lower than that of the fixed cw lidar, before fluctuating within the best practice range at higher wind speeds. Above $15.5\,ms^{-}1$, the MBE turns negative, following the same trend as

the fixed cw lidar. While MBE provides insight into absolute bias, it does not fully capture relative errors, particularly at low wind speeds, where small absolute differences may result in large relative deviations. To address this, we examine the MRBE presented in Figure 11.

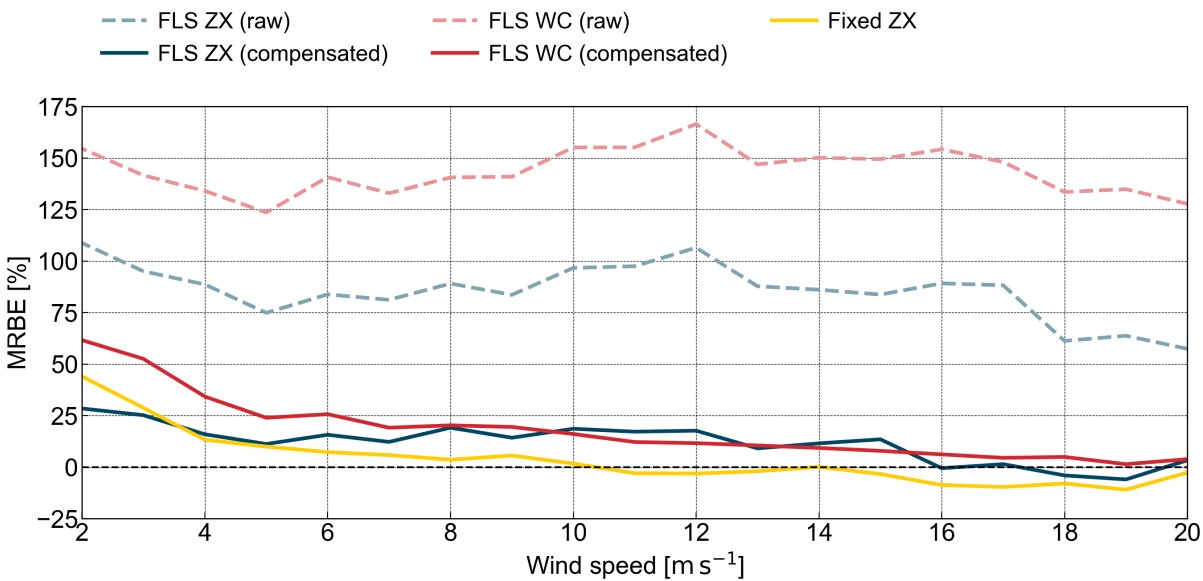

**Figure 11.** Mean Relative Bias Error between the MM TI and the TI of the trialed devices Fixed ZX (yellow), FLS ZX (blue lines, where the dashed line represents raw data and the solid line depicts motion-compensated data) and the FLS WC (red lines, where the dashed line represents raw data and the solid line depicts motion-compensated data) from 101 m above LAT.

The MRBE trends in Figure 11 provide a complementary perspective to MBE. The raw datasets show extremely high MRBE values across all wind speeds, with the floating pulsed lidar reaching from about 125% to 174% and the floating cw

lidar spanning from 60% to 101%. The fixed cw lidar MRBE remains stable across all wind speed bins, indicating that most of the bias is due to platform motion. Following motion compensation, the floating pulsed lidar again experiences the largest relative improvement, with an almost linear decline with increasing wind speeds. At higher wind speeds (betwee $10\,ms^{-}1$ and $15.5\,ms^{-}1$), the compensated pulsed lidar slightly outperforms the floating cw lidar while keeping a poitive bias. This aligns with previous MBE findings that motion compensation is particularly effective for the pulsed floating lidar at higher wind

speeds. For the floating cw lidar, the MRBE closely aligns with that of the fixed cw lidar at lower wind speeds before diverging




around $5\,ms^{-}1$. While MBE is effectively reduced at wind speeds between 10 and $15\,ms^{-}1$, MRBE remains slightly elevated compared to the floating pulsed lidar.

### 3.3 Root Mean Square Error and Relative Root Mean Square Error

The RMSE trends in Figure 12 illustrate the magnitude of absolute errors in TI measurements. The figure follows the same
approach as Figure 10 and 11, showing the error as a function of binned wind speed:

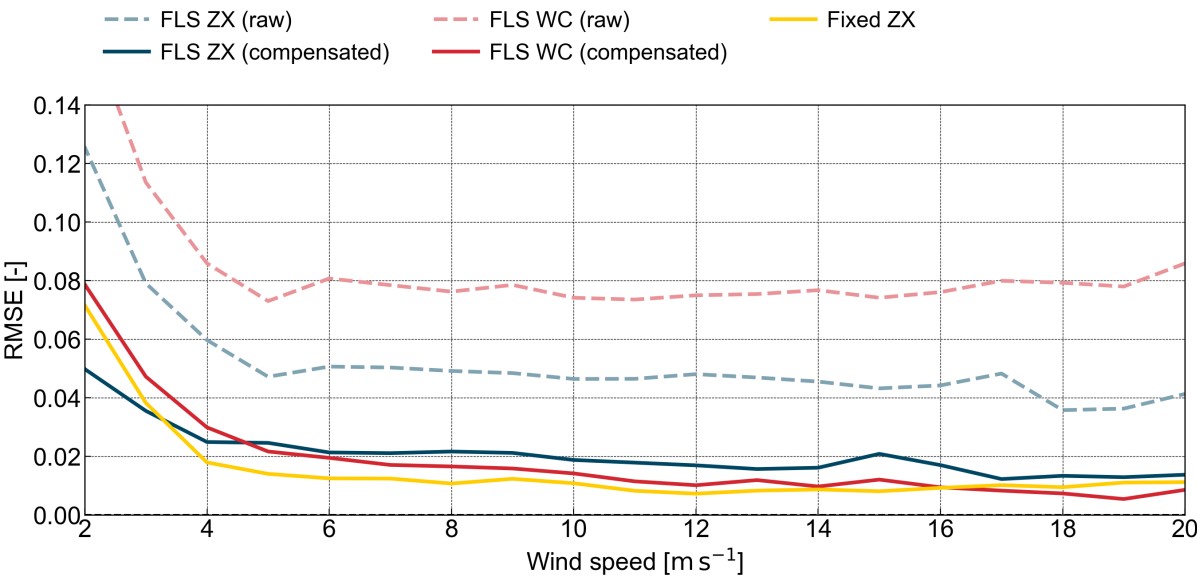

**Figure 12.** Binned Root Mean Square Error between the MM TI and the TI of the trialed devices Fixed ZX (yellow), FLS ZX (blue lines, where the dashed line represents raw data and the solid line depicts motion-compensated data) and the FLS WC (red lines, where the dashed line represents raw data and the solid line depicts motion-compensated data) from 101 m above LAT.

Aligning with the trends from the MBE and MRBE analysis, the raw FLS datasets (dashed lines) exhibit substantially higher RMSE values compared to the fixed cw lidar. The highest RMSE values occur at lower wind speeds, gradually decreasing with increasing wind speed. While both lidar types exhibit high RMSE values in their raw datasets, the floating pulsed lidar (red dashed line) consistently shows greater RMSE than the floating cw lidar (blue dashed line). This suggests that, in addition to
the higher systematic bias observed in MBE and MRBE, the motion introduces greater random errors into the pulsed lidar TI.

The fixed cw lidar (yellow line) exhibits a relatively high RMSE at low wind speeds, which steadily declines and stabilizes at a low level beyond $4\,ms^{-}1$. This trend again highlights the differences between TI measurements from cw lidars and those derived from a cup anemometer. Even without motion, RMSE does not reach zero, suggesting that part of the error arises from differences in measurement principles rather than motion alone.
Following motion compensation, RMSE is significantly reduced for both lidar types (blue and red solid lines).





The floating pulsed lidar TI (red solid line) experiences the largest relative improvement in RMSE, following a steep decline at low wind speeds. Notably, for wind speeds above $4.5\,ms^{-}1$, the RMSE in the pulsed lidar TI is lower than that of the floating cw lidar, and for wind speeds above $16\,ms^{-}1$, it even outperforms the fixed cw lidar, suggesting better alignment with the cup anemometer TI in high wind speed conditions. For the floating cw lidar, RMSE after motion compensation is clearly

reduced but remains consistently higher than that of the floating pulsed lidar and fixed cw lidar across all wind speed bins beyond $3.5\,ms^{-}1$. This again indicates that while the compensation is effective, some residual motion effects may persist in the floating cw lidar TI measurements.

While RMSE provides a direct measure of absolute TI deviations, it does not account for how these errors scale with the TI magnitude itself. Since TI varies significantly across different wind speeds, an identical RMSE value at low and high

wind speeds can have different implications for measurement accuracy. To capture this effect, we have analyzed the RRMSE, presented in Figure 13:

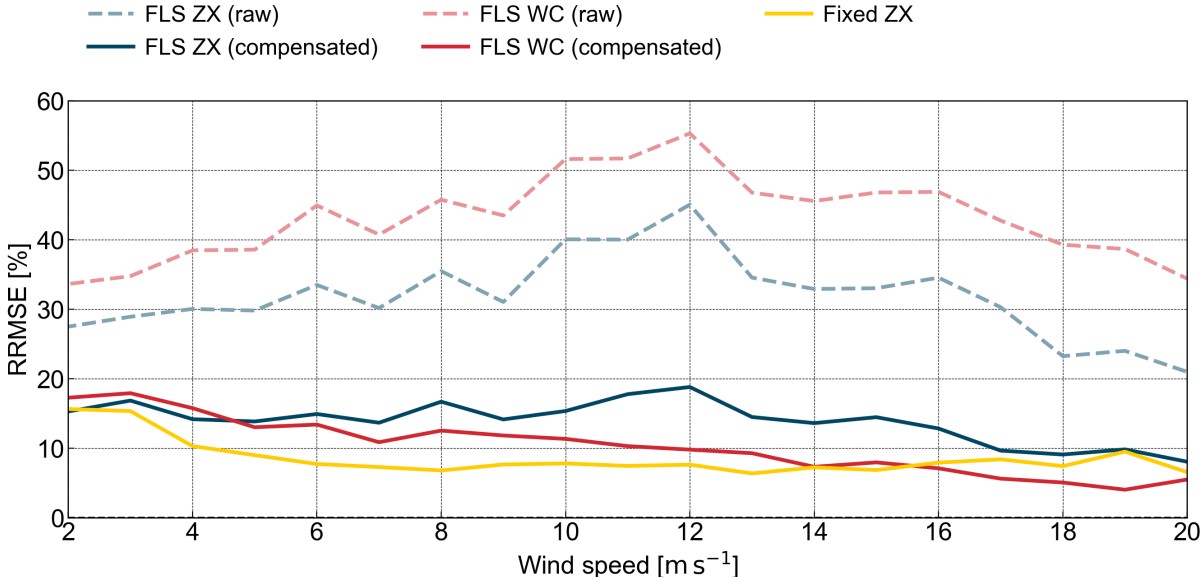

**Figure 13.** Relative Root Mean Square Error between the MM TI and the TI of the trialed devices Fixed ZX (yellow), FLS ZX (blue lines, where the dashed line represents raw data and the solid line depicts motion-compensated data) and the FLS WC (red lines, where the dashed line represents raw data and the solid line depicts motion-compensated data) from 101 m above LAT.

The RRMSE trends in Figure 13 reveal patterns that were less apparent in the RMSE results. While the RMSE decreases with increasing wind speed, the RRMSE remains relatively high across all wind speed bins, with peaks at moderate wind. Those peaks are particularly pronounced in the raw datasets (dashed lines).

The pulsed lidar (red dashed line) shows the highest RRMSE, reaching values of up to 56%, while the floating cw lidar (blue dashed line) exhibits values between 30% and 45%. The fixed cw lidar TI RRMSE remains below 10% across all wind speeds above $5\,ms^{-}1$, indicating the impact of the different measurement principles on the TI RRMSE. Following motion





compensation, the floating pulsed lidar again exhibits the largest relative improvement, with RRMSE decreasing almost linearly as wind speeds increase. At wind speeds above $5\,ms^{-}1$, the compensated pulsed lidar performs better than the floating cw
lidar. Above $14\,ms^{-}1$, the RRMSE of the floating pulsed lidar is even lower than that of the fixed cw lidar, indicating a better alignment of the pulsed lidars TI with cup anemometer TI. While the RMSE of the floating cw lidar decreases with increasing wind speed, the RRMSE fluctuates at lower wind speeds until peaking close around 19% at the $12\,ms^{-}1$ bin. For wind speeds beyond $12\,ms^{-}1$, the RRSME decreases and almost aligns with the fixed cw lidars RRMSE around the $17\,ms^{-}1$ bin.

### 3.4 Representative TI Error

While previous metrics provided valuable insights into bias and variability in the TI measurements, they do not necessarily indicate how well the lidar-derived TI represents statistical reference values for real-world applications (Q90). To assess the overall accuracy of TI estimates, we have analyzed the Representative TI Error as a function of binned wind speed, as shown in Figure 14, while utilizing the same approach as in the previous figures. Additionally, KPIs are indicated in Figure 14, with the black dashed line representing the minimum practice threshold and the green dashed line denoting the best practice range.
The Representative TI Error was calculated according to Equation 8 in Section 2.3.4, while using the Q90 values derived from the TI distributions.

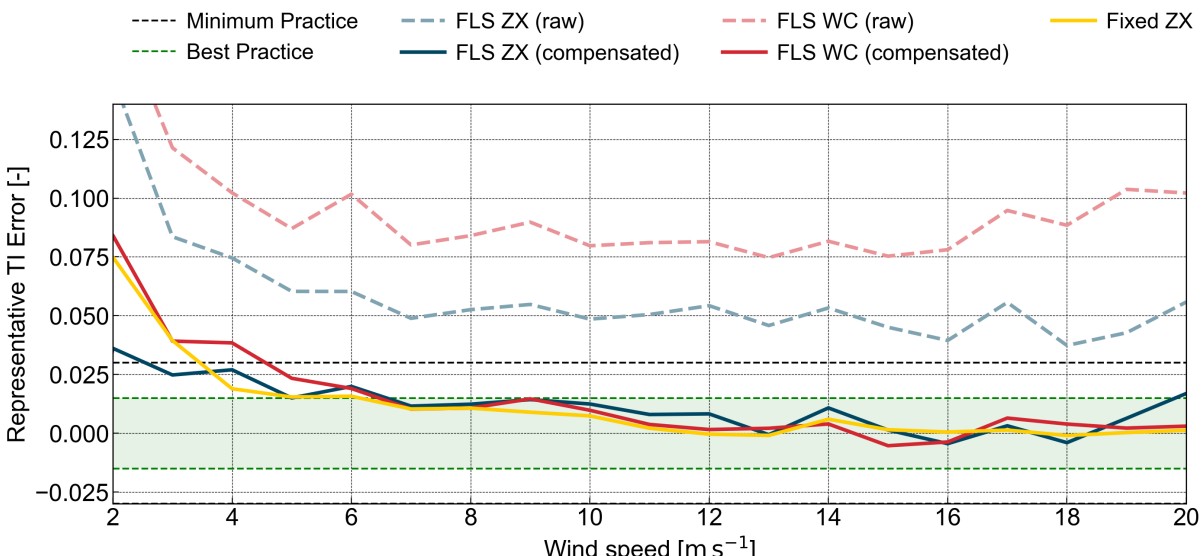

**Figure 14.** Representative TI Error between the MM TI and the TI of the trialed devices Fixed ZX (yellow), FLS ZX (blue lines, where the dashed line represents raw data and the solid line depicts motion-compensated data) and the FLS WC (red lines, where the dashed line represents raw data and the solid line depicts motion-compensated data) from 101 m above LAT.

Similar to previous metrics, the raw floating lidar datasets (red and blue dashed lines) exhibit substantially larger errors compared to its motion-compensated versions and the fixed cw lidar (solid lines).





The floating pulsed lidar (red dashed line) exhibits the highest Representative TI Error, followed by the raw floating cw

lidar (blue dashed line). This is consistent with previous observations. The fixed cw lidar (yellow line) shows an initially high Representative TI Error which gradually decreases with increasing wind speeds. For wind speeds above $3\,ms^{-}1$ it reaches the minimum practice area, before entering the best-practice range for wind speeds above $6\,ms^{-}1$. Beyond the $12\,ms^{-}1$ bin, the Representative TI Error almost aligns with the zero-error line. This suggests that cw lidar TI differs from cup anemometer TI at lower wind speeds. Following motion compensation, the Representative TI Error is significantly reduced for both lidar types.

The floating pulsed lidar (solid red line) shows the greatest relative improvement. Following a steep decline in Representative TI Error at low wind speeds, it almost aligns with the fixed cw lidars trend for wind speeds beyond the $6\,ms^{-}1$ bin. For wind speeds above $14\,ms^{-}1$, the compensated pulsed lidar starts fluctuating around the zero-error line. The motion-compensated floating cw lidar trend (solid blue line) shows the lowest initial Representative TI Error, displaying an even lower error than the fixed cw lidar for wind speeds below $4\,ms^{-}1$. Overall the Representative TI Error of the motion-compensated floating cw

lidar decreases with increasing wind speed before passing the zero-error line at the $13\,ms^{-}1$ bin and fluctuating around it for wind speeds beyond that. However, at moderate wind speeds ($9–15\,ms^{-}1$), the floating cw lidar retains slightly higher error values compared to the pulsed lidar.

## 3.5  Quantile-based distribution analysis

While previous analyses have focused on statistical errors and bias trends, a Q-Q plot provides an alternative way to evaluate

how well the TI distributions align with the reference MM TI. Figure 15 presents a Q-Q plot comparing the floating (both raw and motion-compensated) and fixed lidar TI measurements against the MM TI at 101 m above LAT:





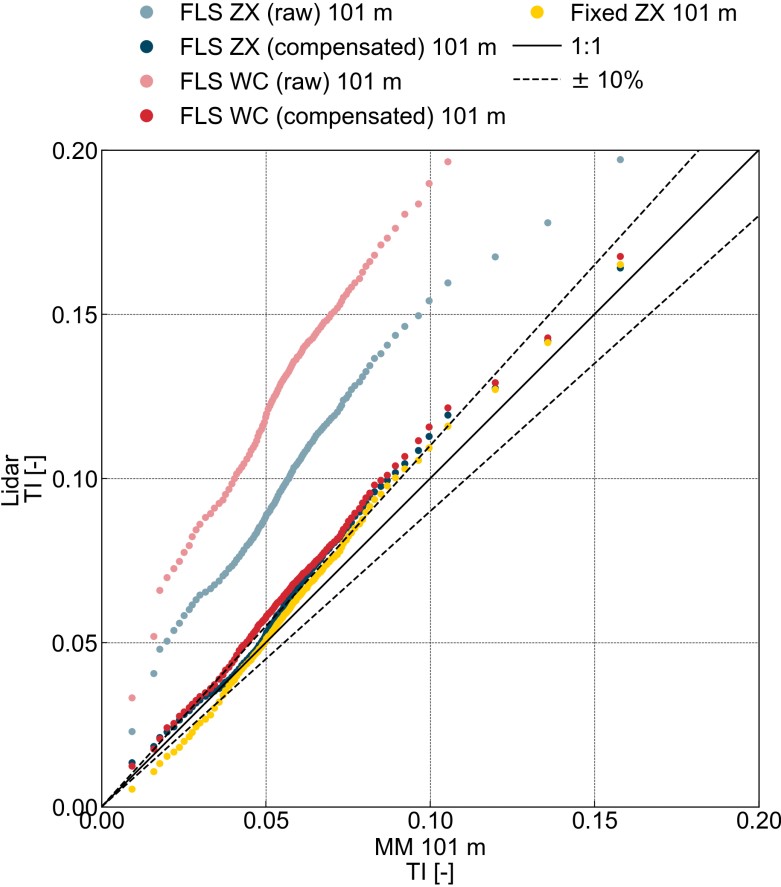

**Figure 15.** Quantile-Quantile plot comparing the quantile distribution between the MM TI and the TI of the trialed devices Fixed ZX (yellow), FLS ZX (blue lines, where the dashed line represents raw data and the solid line depicts motion-compensated data) and the FLS WC (red lines, where the dashed line represents raw data and the solid line depicts motion-compensated data) from 101 m above LAT.

The raw FLS TI datasets (light blue and light red) deviate significantly from the 1:1 line, indicating systematic TI over-estimation compared to met mast cup TI. The raw pulsed lidar (light red) exhibited the largest deviations, with deviations increasing as TI values rise. The raw cw lidar (light blue) displayed a slightly lower offset. The small deviations shown by the fixed cw lidar (yellow), emphasize the difference between lidar and cup TI. Motion compensation significantly improves the agreement between floating lidar TI estimates and the MM reference. The motion-compensated datasets (blue and red dots) shift toward the 1:1 lines. The deviation from the ±10% threshold is substantially reduced, confirming the effectiveness of the applied correction algorithm. The floating pulsed lidar (red dots) sees the most noticeable improvement, with compensated TI falling much closer to the 1:1 line. However, a slight overestimation persists, suggesting a minor systematic bias. The floating cw lidar (blue dots) also shows a strong improvement. For lower TI values, the distance to the 1:1 line is comparable to that





of the fixed cw lidar although its overestimating rather than underestimating. For TI values above 0.04, the floating cw lidar almost aligns with the fixed cw lidar.

## 4    Discussion

The results of this study demonstrate the efficiency of the deterministic motion compensation algorithm in improving the
accuracy of FLS TI measurements. By comparing raw FLS data, motion-compensated FLS data, and fixed cw lidar data against a met mast cup reference, we evaluated the algorithm's impact using multiple error metrics. A key objective was to determine how well motion-compensated FLS data can match the performance of a fixed cw lidar when compared to reference met mast cup TI. To provide a comprehensive assessment, the discussion is structured into two main aspects: accuracy metrics, which assess systematic deviations (bias) between FLS and the reference measurement, and precision metrics, which evaluate
the scatter and consistency of the measurements.

Accuracy was primarily assessed through MBE, MRBE, and Representative TI Error. The raw FLS TI data exhibited a consistent positive MBE, meaning that both cw and pulsed lidars overestimated TI compared to the fixed cw lidar reference. This overestimation was stronger in the pulsed lidar data, likely due to its different spatial and temporal measurement characteristics. After motion compensation, the MBE was significantly reduced for both lidar types, particularly at moderate and
higher wind speeds. The cw lidar achieved the lowest MBE among the compensated datasets, indicating that the algorithm was particularly effective at mitigating systematic bias for this lidar type. The MRBE results confirmed this trend, while revealing lower values for the pulsed lidar at moderate wind speeds. Overall, the analysis showed that the motion compensation effectively reduced relative bias across all wind speed bins. The compensated datasets showed near-zero bias at higher wind speeds, further validating the motion correction approach. The Representative TI Error analysis, which assesses the error in the 90th
quantile of the TI distribution, a crucial parameter for wind turbine design applications, showed a notable improvement after motion compensation. The compensated datasets closely aligned with the suggested best practice threshold, indicating that the motion correction algorithm successfully minimized systematic errors in TI estimates.

Precision was evaluated through RMSE, RRMSE, correlation analysis ($R^2$ and linear regression), and quantile-based distribution analysis. The RMSE results highlighted the impact of the motion compensation in reducing scatter and improving
measurement stability. The pulsed lidar initially exhibited the highest RMSE among the raw datasets, but after compensation, its RMSE was significantly reduced, surpassing the performance of the compensated cw lidar at higher wind speeds. The RRMSE, which normalizes RMSE, further confirmed this pattern. The motion-compensated pulsed lidar dataset showed the lowest RRMSE at high wind speeds, even outperforming the fixed cw lidar, indicating that the pulsed lidar TI aligns closer with a cup TI at higher wind speeds than a cw lidar TI.

Linear regression and correlation analysis demonstrated the overall improvement in agreement between FLS and reference data. The motion compensation notably increased the $R^2$ for both lidar types, with the pulsed lidar experiencing the largest relative improvement. The remaining scatter can be attributed to the distance between the devices, the elevation of the fixed cw lidar, motion effects not covered by the algorithm, small time offsets between lidar and IMU data, height variations due





to heave and tilt, and differences in measurement principles between the lidars and cup anemometers. The quantile-based
distribution analysis (Q-Q plot) provided further insight into systematic over- and underestimation. The motion-compensated
datasets exhibited better alignment with the reference data, reducing deviations across the distribution.

This study confirms that deterministic motion compensation significantly improves the accuracy and precision of FLS-
derived TI measurements. While both lidar types benefited from the algorithm, the pulsed lidar exhibited a greater reduction in
RMSE and RRMSE, whereas the cw lidar achieved better bias correction (lower MBE and MRBE in most bins). The results
from the Representative TI Error analysis further validate the efficiency of the FLS motion compensation algorithm. The
error of the resulting motion-compensated datasets closely align with that of the fixed cw lidar. As both lidars were mounted
on FLS of the same type (Fraunhofer IWES Wind Lidar Buoy) and only timestamps where all systems recorded data were
considered, the observed differences must be caused by the different underlying measurement principles and the lidar system's
response to platform motion. The systematic TI overestimation in the raw floating pulsed lidar data was consistently higher
than that observed in the raw floating cw lidar measurements. This discrepancy is likely caused by the DBS scanning method
of the pulsed lidar, where each LoS is measured sequentially at different azimuth angles, making it more sensitive to short-
term platform motions. Although the deployed pulsed lidar was configured to collect data at an accelerated frequency of 5 Hz
(compared to the 1 Hz standard configuration), the system was clearly more sensitive to motion than the cw lidar. A pulsed
lidar of the same type with a lower scan frequency would likely result in an even higher overestimation. In contrast, the cw
lidar performs continuous conical scans, averaging the measured LoS velocity and thereby already reducing the influence of
rapid motions. The continuous scan with the 50 unsigned LoS measurements however appears to be more sensitive to small
time offsets, which may lead to residual motion effects.

Despite the success of the motion compensation algorithm, some sources of residual error remain (e.g. the remaining scatter
in the regression plots and slight deviations in the quantile-based distribution analyses). Future work could focus on refining
the motion compensation algorithm by accounting for further sensitivities such as atmospheric variations, lidar type specific
sensitivities such as the measurement volume or focal length as well as residual motion effects that are not fully captured
by the compensation algorithm. In this context, machine learning models could play a complementary role, enhancing the
motion-compensated data by accounting for complex, non-linear patterns and relationships that are difficult to model deter-
ministically. However, it is essential that the deterministic motion compensation algorithm remains the foundation, providing
a transparent framework. Once established, machine learning can build upon this foundation as an additional layer, offering
further improvements without sacrificing traceability and understanding of the applied corrections.



## 5 Conclusions

This study evaluated the accuracy and precision of FLS TI measurements by comparing raw and motion-compensated FLS TI data, as well as fixed cw lidar TI data, against a met mast cup anemometer TI reference. The results demonstrate that

motion-induced overestimation in raw FLS TI data can be effectively corrected through deterministic motion compensation, significantly reducing systematic bias and scatter.

After motion compensation, both cw and pulsed lidar TI exhibited a strong alignment with TI derived from a fixed cw lidar across all examined error metrics. The cw lidar TI achieved the lowest bias after compensation, while the pulsed lidar TI showed the most significant improvement in scatter-related metrics, such as RMSE and RRMSE. Overall, the pulsed lidar TI

showed the greatest relative improvements.

Overall, the findings confirm that properly motion-compensated FLS can achieve TI measurements comparable to those of a fixed lidar, making them suitable for offshore wind site assessments. Future work should focus on refining the motion compensation algorithm by accounting for lidar sensitivities, improving sensor synchronization, and investigating the performance under different environmental conditions.







**Figure A1.** TI Error between the MM TI and the TI of the trialed devices Fixed ZX (yellow), FLS ZX (blue lines, where the dashed line represents raw data and the solid line depicts motion-compensated data) and the FLS WC (red lines, where the dashed line represents raw data and the solid line depicts motion-compensated data) from 71 m above LAT; (a) mean TI; (b) TI Q90; (c) TI MBE; (d) TI RMSE; (e) TI RMBE; (f) TI RRMSE; (g) Representative TI; (h) count.



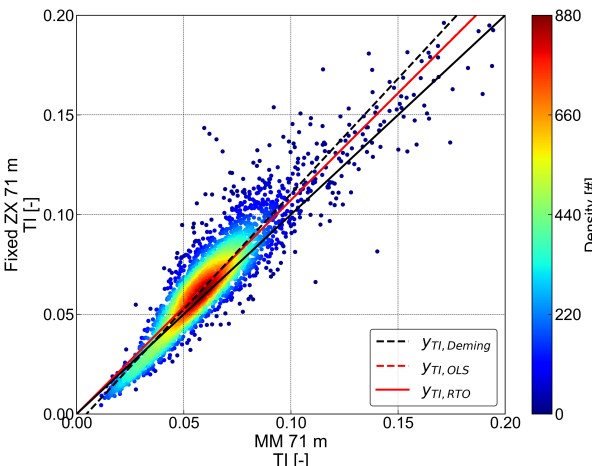

**Figure A2.** Density correlation plot and regression analysis of the fixed cw lidar (Fixed ZX) TI versus the met mast cup anemometer (MM) TI at 71 m above LAT. The point density is indicated by the color bar. Derived parameters are listed in Table A1.

**Table A1.** Regression parameters for the correlation between TI measured by the fixed cw lidar (Fixed ZX) and the met mast cup anemometer (MM) at 71 m above LAT, as illustrated in Figure A2.

|  | **Deming** | **OLS** | **RTO** |
|---|---|---|---|
| **Slope** | 1.157 | 1.074 | 1.073 |
| **Intercept** | -0.005 | -0.000 | - |
| $R^2$ | 0.872 | 0.877 | 0.877 |
| **Number of 10-minute data points** | | 3,021 | |




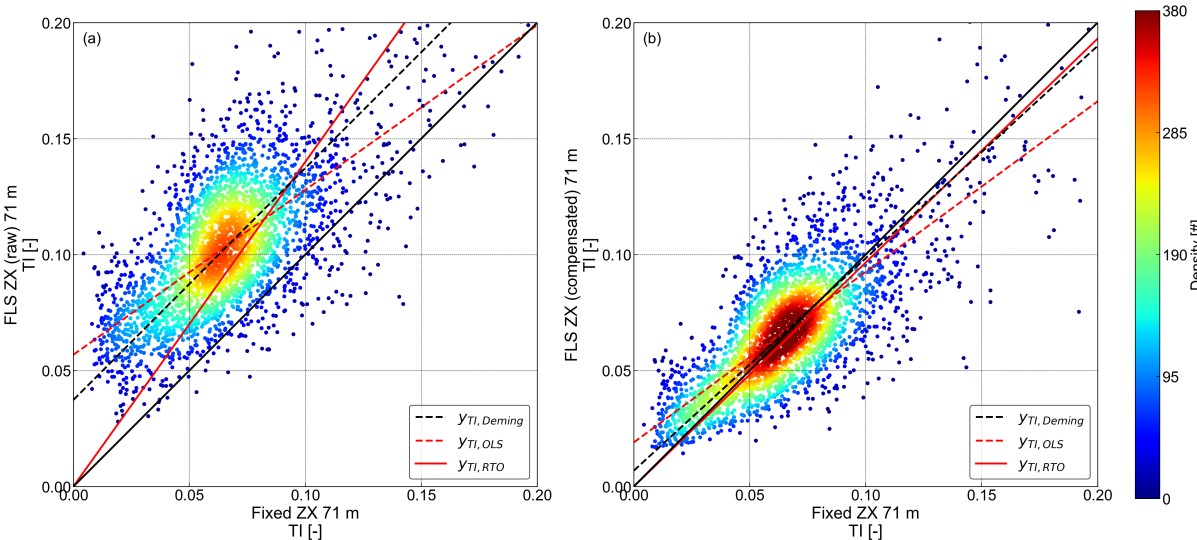

**Figure A3.** Density correlation plots and regression analyses of TI measured by the floating cw lidar (FLS ZX) versus the fixed cw lidar (Fixed ZX) at 71 m above LAT. Panel (a) shows uncompensated (raw) data, while panel (b) shows motion-compensated data. Point density in both panels is indicated by the color bar. Derived parameters are listed in Table A2

**Table A2.** Regression parameters for the correlation between TI measured by the floating cw lidar (FLS ZX) and the fixed cw lidar (Fixed ZX) at 71 m above LAT, as illustrated in Figure A3.

|  | Panel (a): Raw data | | | Panel (b): Motion-compensated data | | |
|---|---|---|---|---|---|---|
|  | **Deming** | **OLS** | **RTO** | **Deming** | **OLS** | **RTO** |
| **Slope** | 0.999 | 0.711 | 1.400 | 0.916 | 0.735 | 0.965 |
| **Intercept** | 0.037 | 0.057 | - | 0.007 | 0.019 | - |
| $R^2$ | 0.422 | 0.505 | NaN | 0.584 | 0.621 | 0.547 |
| **Number of 10-minute data points** | 3,021 | | | 3,021 | | |





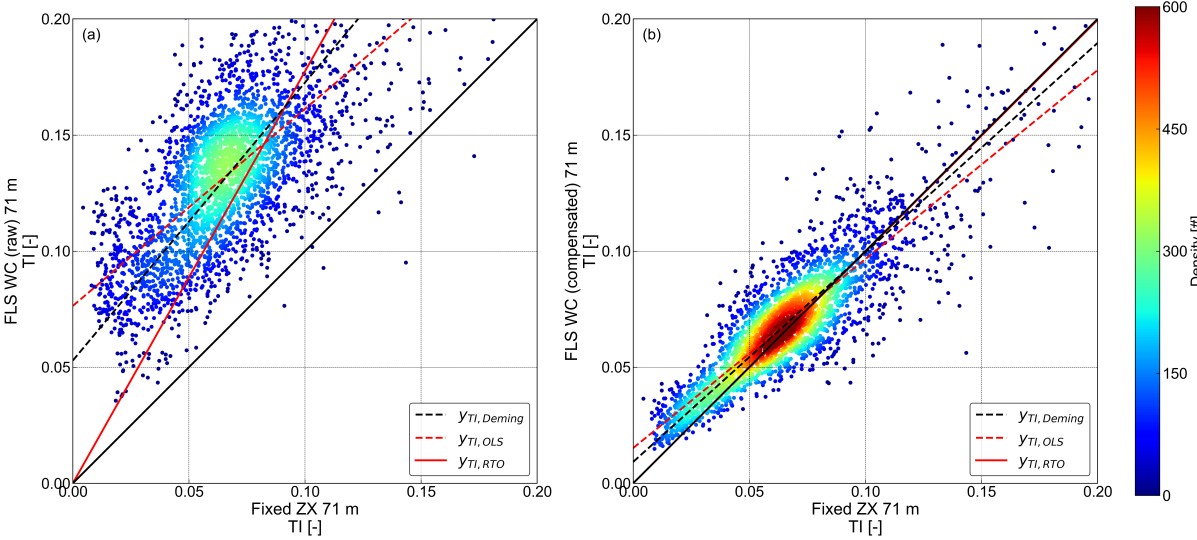

**Figure A4.** Density correlation plots and regression analyses of TI measured by the floating pulsed lidar (FLS WC) versus the fixed cw lidar (Fixed ZX) at 71 m above LAT. Panel (a) shows uncompensated (raw) data, while panel (b) shows motion-compensated data. Point density in both panels is indicated by the color bar. Derived parameters are listed in Table A3

**Table A3.** Regression parameters for the correlation between TI measured by the floating pulsed lidar (FLS WC) and the fixed cw lidar (Fixed ZX) at 71 m above LAT, as illustrated in Figure A4.

|  | Panel (a): Raw data | | | Panel (b): Motion-compensated data | | |
|---|---|---|---|---|---|---|
|  | **Deming** | **OLS** | **RTO** | **Deming** | **OLS** | **RTO** |
| **Slope** | 1.197 | 0.848 | 1.774 | 0.902 | 0.814 | 0.998 |
| **Intercept** | 0.053 | 0.076 | - | 0.009 | 0.015 | - |
| $R^2$ | 0.457 | 0.551 | NaN | 0.787 | 0.796 | 0.746 |
| **Number of 10-minute data points** | 3,021 | | | 3,021 | | |



**Figure A5.** TI error between the MM TI and the TI of the trialed devices Fixed ZX (yellow), FLS ZX (blue lines, where the dashed line represents raw data and the solid line depicts motion-compensated data) and the FLS WC (red lines, where the dashed line represents raw data and the solid line depicts motion-compensated data) from 107 m above LAT; (a) mean TI; (b) TI Q90; (c) TI MBE; (d) TI RMSE; (e) TI RMBE; (f) TI RRMSE; (g) Representative TI; (h) count.



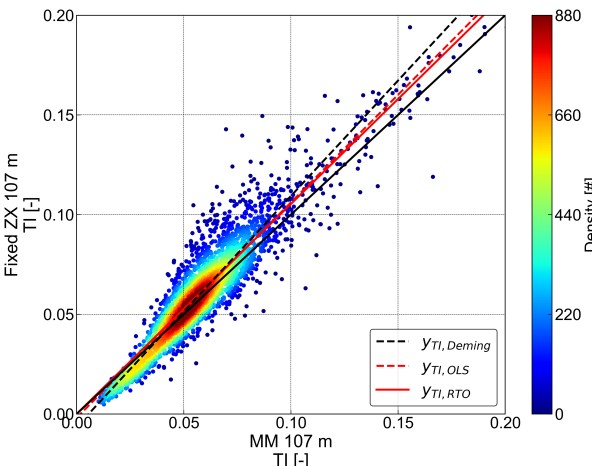

**Figure A6.** Density correlation plot and regression analysis of the fixed cw lidar (Fixed ZX) TI versus the met mast cup anemometer (MM) TI at 107 m above LAT. The point density is indicated by the color bar. Derived parameters are listed in Table A4.

**Table A4.** Regression parameters for the correlation between TI measured by the fixed cw lidar (Fixed ZX) and the met mast cup anemometer (MM) at 107 m above LAT, as illustrated in Figure A6.

|  | **Deming** | **OLS** | **RTO** |
|---|---|---|---|
| **Slope** | 1.158 | 1.080 | 1.051 |
| **Intercept** | -0.007 | -0.002 | - |
| $R^2$ | 0.880 | 0.885 | 0.884 |
| **Number of 10-minute data points** | 3,031 | | |





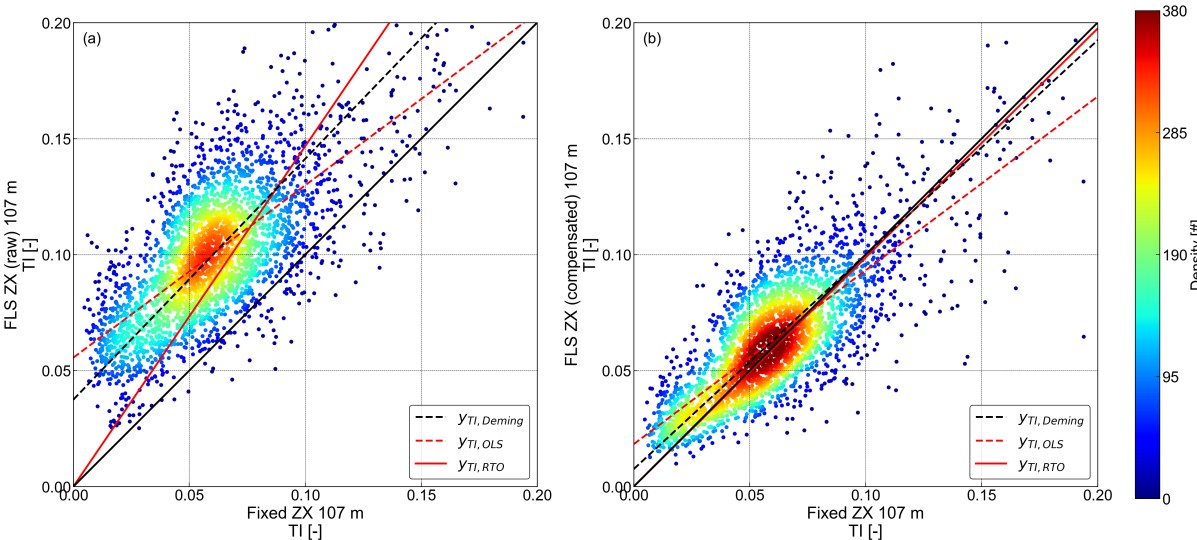

**Figure A7.** Density correlation plots and regression analyses of TI measured by the floating cw lidar (FLS ZX) versus the fixed cw lidar (Fixed ZX) at 107 m above LAT. Panel (a) shows uncompensated (raw) data, while panel (b) shows motion-compensated data. Point density in both panels is indicated by the color bar. Derived parameters are listed in Table A5

**Table A5.** Regression parameters for the correlation between TI measured by the floating cw lidar (FLS ZX)and the fixed cw lidar (Fixed ZX) at 107 m above LAT, as illustrated in Figure A7.

| | Panel (a): Raw data | | | Panel (b): Motion-compensated data | | |
|---|---|---|---|---|---|---|
| | **Deming** | **OLS** | **RTO** | **Deming** | **OLS** | **RTO** |
| **Slope** | 1.044 | 0.745 | 1.464 | 0.933 | 0.758 | 0.994 |
| **Intercept** | 0.037 | 0.055 | - | 0.007 | 0.018 | - |
| $R^2$ | 0.439 | 0.522 | NaN | 0.608 | 0.642 | 0.564 |
| **Number of 10-minute data points** | | 3,031 | | | 3,031 | |





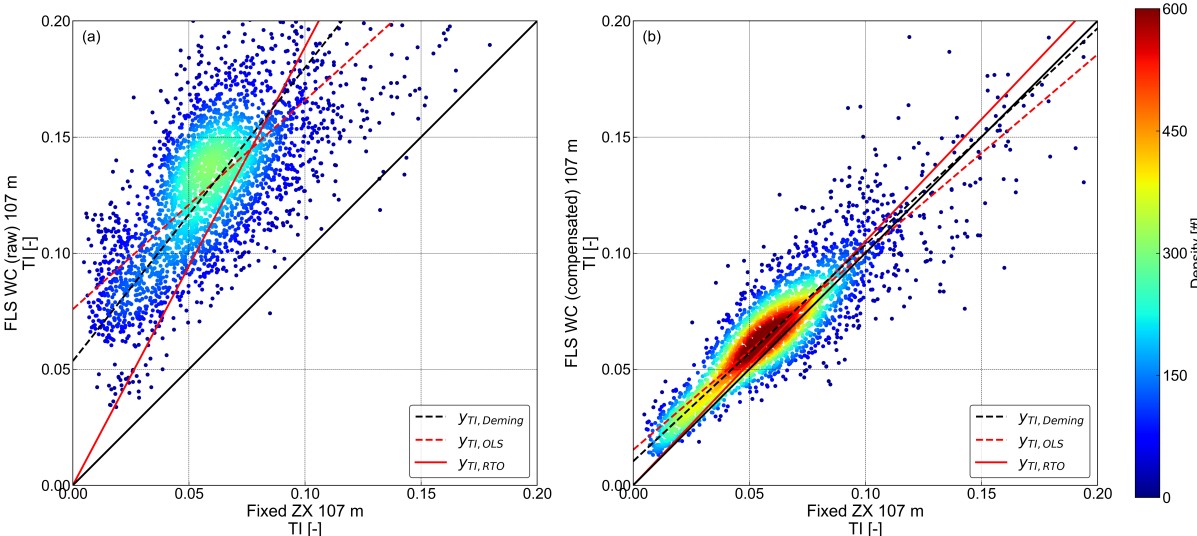

**Figure A8.** Density correlation plots and regression analyses of TI measured by the floating pulsed lidar (FLS WC) versus the fixed cw lidar (Fixed ZX) at 107 m above LAT. Panel (a) shows uncompensated (raw) data, while panel (b) shows motion-compensated data. Point density in both panels is indicated by the color bar. Derived parameters are listed in Table A6

**Table A6.** Regression parameters for the correlation between TI measured by the floating pulsed lidar (FLS WC) and the fixed cw lidar (Fixed ZX) at 107 m above LAT, as illustrated in Figure A8.

| | Panel (a): Raw data | | | Panel (b): Motion-compensated data | | |
|---|---|---|---|---|---|---|
| | **Deming** | **OLS** | **RTO** | **Deming** | **OLS** | **RTO** |
| **Slope** | 1.260 | 0.901 | 1.881 | 0.933 | 0.853 | 1.048 |
| **Intercept** | 0.053 | 0.076 | - | 0.010 | 0.015 | - |
| $R^2$ | 0.481 | 0.571 | NaN | 0.817 | 0.824 | 0.771 |
| **Number of 10-minute data points** | 3,031 | | | 3,031 | | |

## A.1

*Author contributions.*

WW developed and implemented the methods, processed and analyzed the data, and drafted the initial manuscript. GW contributed as a discussion partner during the research process. JG supervised the study and supported the literature review. While WW conducted the primary work, all authors contributed to reviewing the manuscript. All authors have approved the content and agreed to be held accountable for it.






*Competing interests.*

At least one of the (co-)authors is a member of the editorial board of Wind Energy Science. The authors have no other competing interests to declare.


*Acknowledgements.* We thank the Fraunhofer IWES Hangar team for their support in providing, maintaining, and deploying the FLS used in this study. We also acknowledge the NEMO project for funding this research (within the FhG ICON programme). This research is part of our ongoing efforts in wind resource assessment, and we offer commercial measurement services for similar applications. The met mast and wave radac reference datasets were collected and made freely accessible by the BSH Marine Environmental Monitoring Network (MARNET), the
RAVE project (www.rave-offshore.de), the FINO project (www.fino-offshore.de), and cooperation partners of the BSH. Additionally, GPT-assisted writing tools were used to improve the clarity and structure of the manuscript



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
