# Peer review of "Evaluating the Impact of Motion Compensation on Turbulence Intensity Measurements from Continuous-Wave and Pulsed Floating Lidars"

_Wind Energy Science, 2025_

## Author Comment (AC1)

**"Evaluating the Impact of Motion Compensation on Turbulence Intensity Measurements from Continuous-Wave and Pulsed Floating Lidars"**

Revision 1

Warren Watson (warren.watson@iwes.fraunhofer.de), Gerrit Wolken-Möhlmann, and Julia Gottschall

**Authors response to Anonymous Referee #1 comment RC1 (2025-04-10 https://doi.org/10.5194/wes-2025-45-RC1)**

We would like to sincerely thank Anonymous Referee #1 for their detailed, constructive, and thoughtful review of our manuscript. We greatly appreciate the time and effort you invested in evaluating our work. Your comments and suggestions helped us to significantly improve the clarity, structure, and scientific depth of the paper. We have carefully addressed all points raised, revising the manuscript accordingly and providing clarifications where necessary.

Below, we respond to each comment in a point-by-point manner. Reviewer comments are shown in bold, followed by our response and a description of the changes made to the manuscript. Further, the results from the raw Windcube V2.1 pulsed lidar data changed slightly, as the system had been partially applying vector averaging. To ensure consistency, the data was reprocessed using scalar averaging.

**Major Comments**

• Redundancy in Structure: The manuscript could benefit from reducing repetitive explanations, particularly in the outline sections (e.g., Lines 117–128).

We have reviewed the manuscript and revised the introductory paragraphs of all major subsections to remove structural repetition. These have been rephrased to focus on the scientific content rather than reiterating the document outline.

• Methodology Motivation: The paper lacks a clear motivation and novelty statement regarding the proposed motion compensation method. It is not evident how this approach differs from or improves upon existing techniques.

We appreciate this comment and have clarified the motivation and contribution of our work in the revised manuscript, particularly in the introduction (lines 111 - 126), a reworked Section 2.2 and an expanded Discussion (lines 571 - 576).

The novelty of our work lies not in the invention of a new motion compensation algorithm, but in the controlled and transparent application of an established deterministic, geometric method to systematically assess its performance across two different lidar types. Previous studies have typically evaluated motion compensation on single lidar platforms, often under varying environmental conditions and using proprietary or only partially disclosed algorithms. In contrast, our study isolates the impact of lidar type on turbulence intensity measurements by holding both the platform motion and the compensation algorithm constant.

The applied algorithm, originally developed by one of the authors and described in Wolken-Möhlmann et al. 2014, has been further refined and is fully disclosed in this study. By deploying a cw and a pulsed lidar on floating platforms of the same type and compensating both with the same algorithm under similar offshore conditions, we create a unique experimental configuration. This setup enables a direct comparison of the systems' responses to motion and its compensation, an aspect that has not been

previously documented in the literature.

The multi-metric approach allows for a comprehensive and differentiated evaluation, capturing both systematic and random errors, and enabling meaningful comparison with existing studies and best-practice criteria in the field.

• Description of Lidar Retrieval Methods: The VAD (Velocity Azimuth Display) and DBS (Doppler Beam Swinging) methods should be more thoroughly described. Consider including detailed explanations in an appendix if necessary.

Both lidar retrieval methods were mentioned in the context of their respective systems (Section 2.2.2), we agree with the reviewer that providing a dedicated explanation improves clarity. We have therefore added Appendix A, which briefly summarizes the principles, assumptions, and references for both methods. The main text in Section 2.2 (lines 176 – 177) now includes a reference to this appendix.

• Motion compensation method definition: The manuscript should thoroughly define/formulate the motion compensation method used.

We appreciate this comment and agree that a clearer and more thorough formulation was needed. We have comprehensively revised Section 2.2, which now includes a step-by-step mathematical formulation of the deterministic motion compensation method used.

• Comparison with Existing Literature: The results should be compared to existing studies in the field, such as those by Kelberlau et al. (2020, 2023) and Gutiérrez-Antuñano et al. (2018), which report statistical indicators with similar magnitudes. Such comparisons are essential to contextualize the study's contributions.

We sincerely thank the reviewer for this valuable suggestion. We fully agree that situating our findings in the context of existing literature is essential to highlight the study's contributions and relevance. Accordingly, we have expanded the discussion section (lines 619 – 661) to include detailed comparisons with relevant recent studies, including Kelberlau et al. (2023), Rapisardi et al. (2024), and Uchiyama et al. (2024). These works employ a variety of motion compensation approaches (deterministic, machine-learning-based), and multi-platform comparisons and report statistical indicators that align well with those used in our analysis (e.g., MBE, RMSE, Representative TI, regression slope, and R2). This allowed us to benchmark our results and identify common trends as well as key differences. We also relate these comparisons back to the three core factors identified in our study (lidar type, platform motion, and compensation method), and reflect on how differing sea states, motion characteristics, and reference instrumentation may account for variations in performance across studies.

**Minor Comments**

Sect. 2.

• Figure 1. Typically IMUs use an inertial reference system with respect to North-East-Down (or different). Please indicate those in the figure.

We thank the reviewer for pointing this out. Figure 1 has been updated to include the North-East-Down (NED) reference system for clarity. The figure caption has also been revised to explicitly describe the orientation of the IMU in relation to the NED frame.

**Sect. 3.**

**• Why only use wind speed data within the range 4-16 m/s?**

The wind speed interval of 4–16 m/s was chosen to reflect the typical operational range of modern wind turbines, where turbulence intensity has the greatest impact on turbine performance, particularly up to rated wind speeds around 10 m/s. This range ensures a relevant assessment of TI under realistic operating conditions. Additionally, it aligns with common practices and recommendations in existing guidelines such as the OWA Roadmap 2018 and the IEC 61400 series, which often define or analyse performance metrics within similar wind speed intervals. That said, we note that the results remain very similar even without applying this wind speed filter.

**• A comparison between mean horizontal wind speeds measured by the FLSs and the anemometers should be provided.**

We thank the reviewer for this suggestion. In response, we have added correlation plots comparing the mean horizontal wind speeds from the non-corrected and motion-compensated FLS data with those from the reference sensors in Appendix B (lines 695 and on), and referenced this addition in the Results section of the manuscript. The comparison shows very good agreement across all evaluated systems. The raw FLS data already meets the best-practice criteria for mean wind speed accuracy as defined in the OWA Roadmap 2018. Nevertheless, the application of motion compensation further improves the agreement in respect to regression slope and R2.

 Lines 315-319: This is not expected. Typically, CW lidars measure lower turbulence values than anemometers due to their inherent temporal and spatial averaging. For instance, see https://wes.copernicus.org/articles/10/83/2025/wes-10-83-2025.html and https://journals.ametsoc.org/view/journals/atot/28/7/jtech-d-10-05004\_1.xml . Please, comment on that.

We thank the reviewer for this insightful observation. Indeed, it is generally expected that cw lidars report lower turbulence intensity than cup anemometers due to their inherent spatial and temporal averaging, as demonstrated in the cited studies.

We have re-checked the data and confirmed the finding. We have observed a similar behaviour in other offshore datasets involving fixed cw as well as pulsed lidars at FINO3. We believe this may be attributed to a combination of factors specific to the offshore environment and site configuration, such as atmospheric stability conditions or mast effects. Further, lidar based turbulence measurements suffer from systematic errors caused by inter- (cross contamination) and intra-beam effects which could lead to under- or overestimation (refer to https://wes.copernicus.org/preprints/wes-2024-93/). It is also worth noting that the referenced publications primarily analyse onshore measurement setups, where terrain-induced turbulence and mast wake effects may differ from those in offshore environments. While the typical expectation remains valid in general, our results emphasize the importance of site-specific evaluation when interpreting TI comparisons across different measurement technologies.

• Lines 321-322: Please comment on how the underlying measurement principles of the instruments generate the discrepancies. Do they cause the CW lidar TI over-estimation as well?

The measurement principle for profiling lidar is complex from a geometrical point of view. Under the assumption of temporal and spatial homogeneity of the wind field in the scales of the lidar

measurement, virtual wind vectors are reconstructed. These vectors may deviate from the actual wind vector over the lidar. We only know that the 10 minute average value of the vectors are a good representation of a reference cup and wind direction measurement. Even if components of the turbulence are decreased due to the size of the measurement volume and the duration of one scanning pattern, the method of wind field reconstruction from several radial velocity measurements may result in variations in the reconstructed wind vectors. This research question remains to be investigated further.

---

## Author Comment (AC2)

**"Evaluating the Impact of Motion Compensation on Turbulence Intensity Measurements from Continuous-Wave and Pulsed Floating Lidars"**

Revision 1

Warren Watson (warren.watson@iwes.fraunhofer.de), Gerrit Wolken-Möhlmann, and Julia Gottschall

**Authors response to Anonymous Referee #2 comment RC2 (2025-04-10 https://doi.org/10.5194/wes-2025-45-RC2)**

We sincerely thank Anonymous Referee #2 for their detailed and thoughtful review. We appreciate the considerable time and care dedicated not only to evaluating the scientific content but also to identifying smaller issues such as typographical errors, unclear phrasings, and formatting inconsistencies. The reviewer's comprehensive and constructive feedback has contributed significantly to improving the clarity, accuracy, and overall quality of the manuscript.

Below, we provide a point-by-point response to each of the comments. Reviewer comments are shown in bold, followed by our response and a description of the changes made to the manuscript. Further, the results from the raw Windcube V2.1 pulsed lidar data changed slightly, as the system had been partially applying vector averaging. To ensure consistency, the data was reprocessed using scalar averaging.

**Overall,**

1. More details are needed regarding the floating lidar motion compensation. The article from Wolken-Mohlmann is referenced, but at least a brief description of the method is required. Furthermore, it is not explained clearly what the added value of the applied method is in comparison with already published motion correction methods. In the lines 78 – 80 it is written that the method is adopted "because of the transparency, robustness, as well as versatility of a physics-based correction model". It is hinted this way that there is a lack of those features in the already applied approaches. I think that this part should be discussed more in the introduction since it will explain what the novelty of this study is.

We appreciate this comment and have clarified the motivation and contribution of our work in the revised manuscript, particularly in the introduction (lines 111 - 126), a reworked Section 2.2 and an expanded Discussion (lines 571 - 576).

The novelty of our work lies not in the invention of a new motion compensation algorithm, but in the controlled and transparent application of an established deterministic, geometric method to systematically assess its performance across two different lidar types. Previous studies have typically evaluated motion compensation on single lidar platforms, often under varying environmental conditions and using proprietary or only partially disclosed algorithms. In contrast, our study isolates the impact of lidar type on turbulence intensity measurements by holding both the platform motion and the compensation algorithm constant.

The applied algorithm, originally developed by one of the authors and described in Wolken-Möhlmann et al. 2014, has been further refined and is fully disclosed in this study. By deploying a cw and a pulsed lidar on floating platforms of the same type and compensating both with the same algorithm under similar offshore conditions, we create a unique experimental configuration. This setup enables a direct comparison of the systems' responses to motion and its compensation, an aspect that has not been previously documented in the literature.

The multi-metric approach allows for a comprehensive and differentiated evaluation, capturing both systematic and random errors, and enabling meaningful comparison with existing studies and best-practice criteria in the field.

2. In the manuscript a series of different statistical parameters are estimated and the performance of the two wind lidars is assessed in relation those. The authors are very thorough in explaining the reasoning of testing those parameters. However, there is no conclusion regarding the result. For example, are the results good enough for a wind resource assessment? And what is the improvement in comparison to results of already published studies? In the discussion part it is stated that "some sources of residual error remain". How important are those? and what is the further improvement that can be achieved for using machine leaning methods given the theoretical limitations of measuring atmospheric turbulence using vertical profiling wind lidars?

We thank the referee for raising this comprehensive and insightful set of questions. From the comment, we identify four central aspects that deserve a dedicated response:

**Are the results good enough for wind resource assessment?**

While TI data are not typically used in wind resource assessment (which focuses primarily on wind speed), they are highly relevant for site assessment, wind turbine class selection, and load simulations. In this context, the results meet or approach the performance of a fixed lidar system as well as the best practice thresholds proposed by Kelberlau et al. 2023. This indicates that the motion-compensated TI measurements are of sufficient quality for use in site assessment applications.

**How do the results compare to previously published studies?**

We revised the discussion section (lines 619 – 661) to include comparisons to relevant studies (Kelberlau et al., 2023; Rapisardi et al., 2024; Uchiyama et al., 2024), as also suggested in another referee comment. The results of our deterministic motion compensation algorithm fall within the same magnitude range of key metrics (e.g., MBE, RMSE, Q90) as those reported in the literature. Importantly, our study contributes novel empirical evidence by isolating two of the three main contributors to TI overestimation, platform motion and compensation method, thereby enabling a direct comparison of lidar types under controlled and matched conditions. This aspect adds significant value and differentiates our study from earlier work.

**How important are the remaining residual errors?**

We believe that the remaining discrepancies, mainly seen in scatter-related metrics, are relatively minor and primarily attributable to lidar-specific sensitivities, such as probe volume, temporal resolution, or TI itself. These residuals do not significantly affect the overall data quality, especially when viewed in the context of the assessed metrics, where the performance of a fixed system is widely met or approached.

**What further improvement could be achieved using machine learning?**

While deterministic methods provide transparent, physics-based compensation and can be applied generally across sites, they are inherently limited in capturing non-linear interactions or lidar-specific

sensitivities. Machine learning models could complement the deterministic compensation by learning residual patterns or dependencies not fully captured by physical modelling, potentially improving agreement with fixed lidar or even cup-derived TI. However, we emphasize that ML should build upon the deterministic foundation, ensuring traceability and avoiding the "black-box" nature of purely data-driven models.

3. Regarding the errors in the estimated TI values from the floating wind lidar, was it possible to correlate those with the significant height or frequency of the waves and/or to atmospheric stability?

We investigated potential dependencies of the residual TI errors on sea state parameters, specifically Hs and Tp. After motion compensation, the remaining errors (such as MBE and Representative TI Error) closely followed the trend of the fixed reference system across different sea states. The overall magnitude of the residual errors was low, and we did not observe any strong or systematic sensitivity to specific Hs or Tp values. Given the limited influence and in consideration of the manuscript's length and scope, we opted not to include the detailed analysis in the final version. A more in-depth motion-specific evaluation may be addressed in future work.

**4. I suggest adding in an appendix or at least mention in the manuscript statistical results of the comparison of the mean wind speed estimations between the non-corrected / corrected and the reference sensors.**

We thank the reviewer for this suggestion. In response, we have added correlation plots comparing the mean horizontal wind speeds from the non-corrected and motion-compensated FLS data with those from the reference sensors in Appendix B (lines 695 and on), and referenced this addition in the Results section of the manuscript (line 356). The comparison shows very good agreement across all evaluated systems. The raw FLS data already meets the best-practice criteria for mean wind speed accuracy as defined in the OWA Roadmap 2018. Nevertheless, the application of motion compensation further improves the agreement in respect to regression slope and R2.

5. The analysis of this study focusses on two specific lidar models, not necessarily two wind lidar techniques (i.e. cw and pulsed). For example, two different pulsed lidars could have different performance due to variations among others in the sampling rate, scanning angles, or optical components. I suggest clarifying in the manuscript that the results presented are relative to the ZX300 and the Windcube wind lidars.

We have clarified in multiple parts of the manuscript, e.g. the Abstract (lines 6-7), Introduction (lines 112-113), Section 2.1 (line 144) and Section 2.4 (line 319 – 320) that, while the findings offer general insights into the behavior of cw and pulsed lidar systems, the conclusions drawn in this study are specifically based on the performance of the ZX Lidars ZX300 and the Vaisala WindCube v2.1 systems used for our measurements.

**Specific comments**

**1.** Line 42. I suggest reformulating the beginning of the sentence to clarify that the statement concerns the mean wind speed and direction.

We have revised the sentence (line 42) to explicitly clarify that the statement refers to mean wind speed and direction.

**2. Line 46. I suggest replacing the word "fluctuations" with "errors"**

While we thank the reviewer for this suggestion, we retain the term "fluctuations" in this context, as it more precisely reflects the physical nature of the observed variations in the lidar derived wind speed data, which are induced by platform motion. These are not directly measurement errors but actual changes in the measured line-of-sight velocities due to the relative motion of the lidar system and the changed measurement geometry. Using "errors" could imply an instrumental or processing fault, which is not the case here.

**3. Line 62. It is written "However, they may struggle when faced with complex, nonlinear motions ... " can you please elaborate more about what is meant with this statement?**

To clarify this point, we have revised the sentence (lines 62 - 64) in the manuscript to elaborate on the limitations of a deterministic motion compensation under complex motion conditions. The updated text now highlights that deterministic models can be challenged by irregular platform dynamics, especially those caused by rapidly changing sea states, which complicate the reconstruction of LoS velocities. Additionally, we emphasize the technical dependencies such as sensor accuracy and synchronization, which are critical for a compensation.

**4. Line 92. What is the context of the thresholds proposed by Kelberlau et al 2023? Are they relevant to wind resource assessment?**

We thank the reviewer for this important question. The thresholds proposed by Kelberlau et al. 2023 are based on the measurement data presented within that same study and are intended as performance guidance for evaluating the accuracy of FLS-derived turbulence intensity. As the authors state, "Based on the measurement data presented here, we suggest the constant MBE and representative TI error thresholds listed in Table 1 for the performance assessment of FLS." While these thresholds are not part of a formal standard, they serve as a useful benchmark for interpreting FLS performance in site assessment contexts.

**5. Line 149. I suggest removing the word "virtual" and add that both the VAD and the DBS methods assume a horizontal homogeneity of the wind conditions.**

While we retained the term "virtual," we have clarified its context and the underlying assumptions by adding Appendix A. This appendix provides a brief explanation of the VAD and DBS retrieval methods, including their assumption of homogeneous flow. A reference to this appendix has been added to the sentence at the line you indicated.

**6. Please add information of the heights where cup and sonic anemometers are installed (or refer to Table 2).**

We thank the reviewer for this comment and added a reference to Table 2 (line 195), where the measurement heights are provided.

**7. Line 181. The deterministic motion compensation algorithm used in this study considers the line-of-sight velocities. How different is this method compared to the method applied by Yamaguchi and Ishihara 2016, Keleberlau et al 2020?**

From the theoretical background, the here used algorithm is similar to the algorithms described in the resources mentioned above, because it is the mathematical solution of the geometrical problem of

floating lidars. The here used algorithm was already described and simulated by us in 2010, first measurement results were shown in 2014. The challenge here is the implementation which was improved over years and applied to two different lidar systems, which required different detail solutions. Detail solution for Kelberlau et al. 2020 are not known. The paper from Yamaguchi and Ishihara 2016 only describes a simulation, not an implementation.

**8. Line 185. The motion of the FLS does not distort the scanning geometry but rather rotates it in respect to the measured air volume.**

While a rotation may describe the scanning geometry under idealized or instantaneous-acquisition conditions, we believe this underrepresents the physical process involved. In practice, both evaluated lidar types perform sequential LoS measurements over a period of about ~1 second, during which the FLS platform experiences motion. This results in each beam being emitted from a slightly different position and orientation, effectively deforming the actual scanning pattern.

9. Lines 195 – 200. To mitigate errors associated with time offsets between the lidar and the motion sensor a methodology is proposed according to which as a time offset is selected the one that results in the smallest standard deviation of the 10-minute wind speed. If the time offsets are due to a drift between the two data acquisition systems, then I would guess that the time offset is increasing/decreasing linearly. Is this something that can be observed from the results of this analysis?

Indeed, a linear drift in the time offset was observed in the data from the ZX Lidars ZX300M system, consistent with a gradual desynchronization between the lidar and the motion sensor data acquisition systems before resynchronizing. In contrast, the Vaisala Windcube V2.1 data did not exhibit a significant drift. The optimal time offset remained largely stable throughout the campaign.

**10. Table 2. The measurement heights of the wind lidar are measured from the sea level?**

We added clarification that heights are relative to LAT (lowest astronomical tide) and thank the reviewer for this question.

**11. Page 17. Figure 6. I think that it would enhance the understanding of the results presented in this manuscript if this figure presented the statistics of the wind and wave conditions of the selected data set and not of the whole set of the acquired data.**

We thank the reviewer for this very good idea to further improve transparency. In response, Figure 6 has been updated to show the wind and wave conditions of the filtered dataset used in the analysis.

**12. Line 303. What is the reason of presenting the results of the analysis at 71 m and 107 m at the appendix if there are not discussed in the main part of the manuscript?**

The analysis at 71 m and 107 m was initially included to assess the vertical consistency of the motioncompensated TI measurements. The results confirmed that the compensation approach yields consistent performance across different heights. However, as these findings did not introduce additional insights beyond (see lines 360 - 361) what is already presented at the main analysis height and were not discussed in the manuscript, we have removed them from the appendix.

**13. Line 315. According to the results presented in Fig 7 the TI estimation of the "fixed ZX wind lidar" measures higher TI than the mast. How is this explained? Did the authors apply a quality check on the lidar data?**

Apart from the manufacturer-provided quality filters and our applied wind direction and wind speed filters, no additional quality filters were applied to the fixed ZX lidar data. However, we performed several consistency checks and found good agreement with the met mast in terms of wind speed and direction measurements.

It is indeed commonly expected that cw lidars report lower TI values than cup anemometers due to their inherent spatial and temporal averaging. This trend is well supported by prior studies, particularly those based on onshore measurements.

However, after rechecking our dataset, we confirm that in this case, the fixed ZX lidar reported slightly higher TI values than the mast during the evaluated period. We have observed similar behaviour in other offshore datasets involving fixed cw as well as pulsed lidars at FINO3 platform. We believe this may be due to site-specific influences such as atmospheric stability conditions or flow disturbances related to the mast structure and layout. Further, lidar based turbulence measurements suffer from systematic errors caused by inter- (cross contamination) and intra-beam effects which could lead to under- or overestimation (refer to https://wes.copernicus.org/preprints/wes-2024-93/).

We have now added a corresponding note to reflect that, while the general expectation of lower lidarderived TI holds in many contexts, offshore environments may introduce deviations. This underlines the importance of site-specific evaluation when interpreting TI comparisons across measurement technologies.

**14. Line 319. Can you please elaborate more here regarding the several factors that have an impact on the regression analysis.**

We appreciate the referee's comment and have revised the manuscript (lines 377 – 380) to provide a more detailed explanation of the factors influencing regression outcomes.

**15. Lines 347 – 354. Here it is written that in the case of the cw lidar the homodyne detection is a limitation that introduces errors in the case of the low wind speeds. However, in all the results that are presented the cw wind lidar is performing better than the pulsed wind lidar.**

In our results, the motion-compensated cw lidar (ZX Lidars ZX300M) generally performs better than the pulsed lidar across most evaluated metrics. Nevertheless, we chose to highlight the limitation introduced by the homodyne detection scheme, as it constitutes a known source of potential error. Specifically, homodyne detection may lead to incorrect sign resolution of the radial velocity, particularly at low wind speeds. Since the deterministic motion compensation algorithm relies on the correct sign to remove motion-induced errors, this represents a limitation. Although this effect was not prominent in our dataset, its magnitude has not yet been systematically assessed. We therefore believe it is important to acknowledge this limitation as a potential source of residual error.

**16. Line 350. The homodyne detection is not the only way to detect the Doppler shift in cw wind lidars. Please reformulate to clarify that you refer to the ZX wind lidar.**

We thank the referee for this clarification. We have revised the sentence (line 411) in the manuscript to specify that homodyne detection refers to the ZX Lidars ZX300M wind lidar used in this study.

**17. Lines 391 – 392. Please add a reference of the minimum and best practice performance thresholds.**

The reference to the minimum and best practice performance thresholds (Kelberlau et al., 2023) was already included in the manuscript at the point where these thresholds are introduced (lines 95 – 98).

**18. Lines 404 – 405. What is the direction of the booms of the cup anemometers on the mast?**

Only cup anemometers mounted on booms oriented at 345°N were used in the analysis, as stated in Section 2.1.3 (Line 195).

**19. Lines 483 – 485. I think that the lower error that the motion-compensated floating wind lidar exhibits in comparison to the fixed is a numerical artifact rather a result that demonstrates the efficiency of the method. How is it possible that the performance of the motion correction method can provide results better than that of a fixed wind lidar?**

We agree that it is indeed surprising to observe that the motion-compensated floating lidar occasionally outperforms the fixed lidar in terms of certain error metrics. We have now clarified in the manuscript (lines 544 – 545) that this may result from site-specific influences such as mast wake effects on the fixed reference sensors or the smaller scan diameter of the elevated fixed. We thank the reviewer for this critical observation.

**20. Line 532. It is written "The remaining scatter can be attributed to ... the elevation of the fixed cw lidar". Didn't all wind lidars measure at the same height above the sea level?**

All lidar systems were configured to measure at the same nominal height above sea level. However, due to the different installation elevations of the devices, there are geometric differences in the scan volumes. The fixed cw lidar, mounted on the met mast platform at approximately 27 m above LAT, has a smaller probe length and scan circle compared to the lidars installed just above the waterline on the FLS. These geometric differences can introduce slight variations in the sampled atmospheric volume and its representativeness, particularly in turbulent conditions, and may contribute to the remaining scatter observed in the results.

**21. Lines 569 – 570. Isn't this a repetition of the previous sentence?**

The sentence was indeed repetitive. Our intention was to emphasize that while both lidar types showed improved performance after motion compensation, the pulsed lidar exhibited the greatest relative improvement across all evaluated metrics when compared to its own raw data. We have restructured the sentence (lines 680 – 681) to clarify this point and avoid redundancy.

**Minor comments**

**1. Throughout the manuscript there is a typo in the wind speed units. Roman fonts should be used. Please correct it.**

We thank the reviewer for pointing this out. All occurrences of wind speed units have been corrected to use the appropriate roman font formatting throughout the manuscript.

**2. Line 229. Eq(3) Correct the subscript of TI.**

The subscript in Equation (3) has been corrected accordingly.

**3. Line 331. Replace "indicate" with "indicates"**

We have updated the text and replaced "indicate" with "indicates".

**4. Line 346. Replace "suggest" with "show"**

We have made the change and replaced "suggest" with "show".

**5. Line 422. Replace "betwee" with "between"**

Thank you for pointing that out. We have corrected the typo in and replaced "betwee" with "between."

**6. Line 430. Replace ":" with "."**

We have replaced the colon with a period as suggested.

**7. Line 451. Replace ":" with "."**

We have replaced the colon with a period as suggested.

8. Lines 587 – 588: In the Acknowledgements it is written: "This research is part of our ongoing efforts in wind resource assessment, and we offer commercial measurement services for similar applications." Why is this relevant to the Acknowledgements?

We agree with the reviewer's observation. The statement was not relevant to the Acknowledgements section and has therefore been removed. Thank you for pointing this out.